# Determination of Fungi and Multi-Class Mycotoxins in *Camelia sinensis* and Herbal Teas and Dietary Exposure Assessment

**DOI:** 10.3390/toxins12090555

**Published:** 2020-08-29

**Authors:** Ingars Reinholds, Estefanija Bogdanova, Iveta Pugajeva, Laura Alksne, Darta Stalberga, Olga Valcina, Vadims Bartkevics

**Affiliations:** 1Institute of Food Safety, Animal Health and Environment “BIOR”, Riga LV-1076, Latvia; estefanija.bogdanova@bior.lv (E.B.); iveta.pugajeva@bior.lv (I.P.); laura.alksne@bior.lv (L.A.); olga.valcina@bior.lv (O.V.); Vadims.Bartkevics@bior.lv (V.B.); 2Faculty of Chemistry, University of Latvia, Riga LV-1004, Latvia; 3Faculty of Medicine and Health Sciences, Linköping University, SE-581 83 Linköping, Sweden; stalberga.darta@gmail.com

**Keywords:** mycotoxins, *Camellia sinensis* teas, herbal teas, 2D-LC-TOF-MS, MALDI-TOF-MS, fungi, dietary exposure assessment

## Abstract

In this paper, a study of fungal and multi-mycotoxin contamination in 140 *Camellia sinensis* and 26 herbal teas marketed in Latvia is discussed. The analysis was performed using two-dimensional liquid chromatography with time-of-flight mass spectrometry (2D-LC-TOF-MS) and MALDI-TOF-MS. In total, 87% of the tea samples tested positive for 32 fungal species belonging to 17 genera, with the total enumeration of moulds ranging between 1.00 × 10^1^ and 9.00 × 10^4^ CFU g^−1^. Moreover, 42% of the teas (*n* = 70) were contaminated by 1 to 16 mycotoxins, and 37% of these samples were positive for aflatoxins at concentrations ranging between 0.22 and 41.7 µg kg^−1^. Deoxynivalenol (DON) and its derivatives co-occurred in 63% of the tea samples, with their summary concentrations reaching 81.1 to 17,360 µg kg^−1^. Ochratoxin A (OTA), enniatins, and two *Alternaria* toxins were found in 10–37% of the teas at low concentrations. The dietary exposure assessment based on the assumption of a probable full transfer of determined mycotoxins into infusions indicated that the analysed teas are safe for consumers: the probable maximum daily exposure levels to OTA and the combined DON mycotoxins were only 0.88 to 2.05% and 2.50 to 78.9% of the tolerable daily intake levels.

## 1. Introduction

Traditional *Camellia sinensis* (*C. sinensis*) black and green teas, and teas made from medicinal herbs such as peppermint, chamomile, and dog-rose, are consumed daily as the most favoured beverages in Europe and other countries [1,2]. Caffeine-rich teas are commonly imported from the tea-growing regions in China, Japan, Sri Lanka, India, and Kenya, repacked and blended with natural or artificial flavours, dried fruits, and spices, and sold in supermarkets or special tea houses. In the past decade, *C. sinensis* and *C. assamica* varieties, such as oolong (semi-fermented) and Pu-erh (post-fermented) teas, have spread around the world and gained popularity among non-Asian tea users due to their multiple health-protective (antioxidant, antiobesity, antibacterial, and antitumor) properties associated with the high contents of catechins, flavanols, and specific fungal microbiota present in these teas [3,4].

Preharvest and postharvest factors such as environmental issues (humidity, CO_2_, pests, and insect damage); the thermal resistance of mycotoxins during tea drying, processing, and storage; and poor microclimate conditions can result in contamination of the final tea products [2,5].

The production conditions can also have notable impacts on the fungal growth in different tea varieties. For example, the production of the ripened (shu) Pu-erh tea variety from loosened black tea leaves involves wet-piling fermentation, which can enhance the development of moulds and the potential formation of mycotoxins compared to the aging conditions of raw (sheng) Pu-erh tea, which is produced from nonfermented tea leaves [6,7].

Recent studies of commercial black and green teas based on routine laboratory analyses using high-performance liquid chromatography (HPLC) with tandem mass spectrometry (MS/MS) and fluorescence detectors (FLD) indicated high rates of storage-related mycotoxins, such as aflatoxins (AFs) and ochratoxin A (OTA), which were found in dry tea leaves and into their infusions [8,9,10].

Mannani et al. (2020) reported that more than half of all tested green tea samples (59%, *n* = 76) from a Moroccan tea market were positive for aflatoxin B_1_ (AFB_1_) and other AFs [11]. In 29.5% and 9.3% of the cases, the concentrations of AFB_1_ and the sum of aflatoxins B_1_, G_1_, B_2_, and G_2_ exceeded the maximum limits of 5 and 10 µg kg^−1^ set for these mycotoxins in Morocco [11].

Carraturo et al. reported OTA concentrations ranging from <0.01 to 21.49 µg kg^−1^ in 82.5% of the analysed 16 black teas and 16 green teas from Italian tea shops [12]. The authors also demonstrated that OTA was found in 37.5% of black tea infusions and 68.8% of green tea infusions, with contamination levels reaching up to 1.0 ng L^−1^.

Several recent studies also observed *Fusarium* mycotoxins (e.g., deoxynivalenol (DON), enniatins (ENNs A, A_1_, B, B_1_), etc.) in *C. sinensis* and herbal teas and infusions, but exposure studies showed that the concentration levels of these contaminants were too low to pose substantial hazards to consumers [13,14,15,16].

The spread of mycotoxin-related plant illnesses in different geographic regions has led to the need for screening studies of regulated and nonregulated toxins using sensitive methods for their simultaneous quantification in teas and other products [2].

The aim of the present study was to perform an analysis of the fungal and mycotoxin contaminations in caffeine-rich and herbal teas available on the Latvian market, since most of these commercial teas are imported. The microbial quality was evaluated via mould quantity studies and the identification of fungal species using matrix-assisted laser desorption/ionisation time-of-flight mass spectrometry (MALDI-TOF-MS).

A recently developed in-house method of online heart-cutting two-dimensional liquid chromatography with time-of-flight mass spectrometry (2D-LC-TOF-MS) was used for the analysis of mycotoxins produced by *Aspergillus, Alternaria, Fusarium*, and *Penicillium* species [17].

The dietary exposure assessment assumed that the mycotoxin concentrations determined in the raw dry teas are fully extracted into the beverages during infusion (worst-case scenario). The intake of the upper-bound (UB) level and maximum mycotoxin concentrations determined in the dry samples was evaluated for a conventional tea portion of one cup (200 mL of boiling water and 2 g of dry tea), taking into account survey data of the average intakes of fermented, nonfermented, and herbal teas in Latvia [18].

## 2. Results

### 2.1. Mycotoxin Analysis Method

Two-dimensional liquid chromatography has recently attracted attention as an advanced tool for the multi-contaminant analysis of food products [19,20,21]. This separation technique can overcome the common challenges that one-dimensional LC techniques face in separating compounds with physicochemical characteristics, such as polarities and molecular charges; moreover, 2D-LC techniques can improve data reproducibility, chromatographic resolution, and time and resource usage due to the reduced need for extra sample preparation, especially for automated online 2D-LC and 2D-LC combined with high-resolution MS detection [20]. Effluent transfer from the first dimension (1D) column to the 2D column is commonly performed via the comprehensive (LC × LC) or heart-cutting (LC–LC) modes [21]. The comprehensive separation mode is commonly employed by eluting every peak fraction separately from the first dimension (1D) column to the 2D column, which improves the separation of unknown substances [20]. Heart-cutting 2D-LC provides the partial transfer of eluent fractions from the 1D to 2D column, which allows one to decouple both the 1D and 2D run times without any time constraints on the 2D separation, thereby providing a higher peak resolution for multicomponent analysis, and especially improving the resolution for small fraction components [21]; thus, this is why we used the LC–LC mode to analyse mycotoxins in the present study. As indicated in Table 1, this method met the performance requirements [22,23] for mycotoxin identification and quantification.

A mycotoxin-free Pu-erh matrix was used as the blank matrix for validation of the 2D-LC-TOF-MS method. The estimated performance criteria are shown only for the twenty mycotoxins that were determined to be positive in the analysed tea samples. More detailed information on the method of validation is provided in our forthcoming paper dedicated to the development and optimisation of this instrumental method [17].

The applicability of the method was proved by quality control (QC). QC samples were included in every analysis sequence, as well as being used for quantification matrix-matched calibration on a blank sample, that included a mix of *C. sinensis* teas, and an herbal tea blend. The accuracy of QC values for all the compounds ranged between 70% and 124%.

The coefficients of determination (R^2^) were higher than 0.99, which confirmed the good linear correlation between the detector response and mycotoxin concentrations. 

The trueness and precision were evaluated at three concentration levels (i.e., the lowest, median, and highest concentrations in a linear range (Table 1)). The trueness of the method was calculated from matrix-matched calibrations, which were performed in order to compensate the negative effect of matrix components on the analyte recovery. The intra-day relative standard deviations (RSDs) were estimated over a two-day validation period. The estimated mean recoveries (80–120%) and intraday RSDs (5–13%) ensured that the trueness and precision of the method were fully acceptable. The measurement uncertainty (MU) for the individual mycotoxins ranged from 9 to 28%, which was also acceptable.

The calculated levels of the detection and quantification (LODs and LOQs) were comparable with the sensitivities reported in studies based on sample preparation using offline SPE methodologies or dispersive SPE (QuEChERS) approaches, and detection methods that primarily use HPLC-MS/MS and other MS techniques [2,11,12]. The estimated LOQ for AFB_1_ was 1.8 times lower than the LOQ (0.08 µg kg^−1^) reported by another recent study [10]. The overall performance of the method used in the present study was similar or superior to 1D-LC-MS methods recently reported for multi-mycotoxin analysis in teas [11,12].

### 2.2. Microbial Quality Assessment of the Teas

Almost all the tea samples (*n* = 159), with the exception of seven oolong samples, presented at limited amounts just for mycotoxin analysis, were analysed for their mould content and microbial quality.

Water activity (a_w_), as the characteristic factor influencing mould growth in products, was evaluated using a standardised method [24]. The average from the triplicate studies ranged between 0.33 ± 0.02 and 0.68 ± 0.05. A mean a_w_ value of 0.46 ± 0.05 was determined in most of the *C. sinensis* and herbal tea samples, except for four oolong and two green tea samples which had a_w_ > 0.60. Typically, a_w_ values exceeding 0.8 will stimulate the growth of fungal species. Common conditions for the generation of AFB_1_-producing *A. flavus* and *A. parasiticus* include 0.90 to 0.98 a_w_ at 27 °C or 0.90 to 0.94 a_w_ at 35 °C [25]. Higher a_w_ values (commonly >0.99 a_w_ at 25 °C) are needed for the growth of *F. graminearum* and other *Fusarium* species, which produce typical trichothecenes such as deoxynivalenol (DON) and the T-2 toxins present in cereals, as well as other common agricultural products [26]. According to the reported studies, a_w_ < 0.65 contributes to decreasing cellular metabolism and preventing fungal growth under proper storage conditions [27]. From this viewpoint, the teas in the present study should have a low risk of fungal growth under proper storage conditions over the course of their shelf lives.

The total number of the colony-forming units (CFU g^−1^) was determined after five days of incubation at 25 ± 0.2 °C, according to the standardised procedure for the determination of yeast and total fungal contents in the products with a_w_ < 0.95 [28].

In addition, yeasts were also determined in some of the samples and counted when possible. These samples included one oolong tea with a 1.00 × 10^1^ CFU g^−1^ yeast cell count, eight black teas with yeast counts ranging between 1.00 × 10^1^ and 9.30 × 10^2^ CFU g^−1^, one hibiscus tea with a yeast count of 1.10 × 10^3^ CFU g^−1^, eight chamomile samples (mono teas and tea blends) with yeast counts ranging between 2.00 × 10^2^ and 4.90 × 10^3^ CFU g^−1^, and four peppermint tea samples, including one blend with green tea, which all presented yeast quantities that were too high to be properly counted, even at the highest dilution ratio used (1:10,000).

In total, 13% (*n* = 20) of the analysed teas presented no signs of mould growth. These samples included one black tea, six green teas, eleven Pu-erh teas (one raw (sheng) Pu-erh and 10 ripe (shu) Pu-erh samples), one chamomile sample, and one peppermint sample. Most of the tea samples (87%, *n* = 138) presented mould counts of 1.00 × 10^1^–9.00 × 10^4^ CFU g^−1^.

As indicated in Figure 1, higher mould counts were observed in the chamomile and peppermint varieties than in the samples of *C. sinensis* tea samples and other herbal varieties. Additional lines presenting recommended total fungal colony-forming unit counts in teas reported by the Tea and Herbal Infusions Europe (THIE) association and the recommendations of the Regulation No 461 of the Cabinet of Ministers of Republic of Latvia [29] are included in Figure 1.

The mean mould counts in green teas were around 1.09 × 10^3^ CFU g^−1^, which exceeded the mean counts of the moulds determined in black teas (5.10 × 10^2^ CFU g^−1^) by 2.1, 14.4, and 15.8 times, especially Pu-erh teas (7.56 × 10^1^ CFU g^−1^) and oolong teas (6.86 × 10^1^ CFU g^−1^). 

Fourteen of the green teas (mainly pure teas or blends with fruits and peppermint) were identified to have mould counts exceeding 5.00 × 10^2^ CFU g^−1^. Mould counts ranged from 6.00 × 10^1^ to 1.6 × 10^4^ CFU g^−1^ in these samples. Most of these samples were bagged green teas from China, and only three were loose-leaf (e.g., weighable) tea samples. One was tea from Sri Lanka, with 6.60 × 10^2^ CFU g^−1^. The other loose-leaf green tea samples were determined to have low quantities of mould between 1.00 × 10^1^ and 4.20 × 10^2^ CFU g^−1^. The lowest fungal mould levels of 1.00–3.00 × 10^1^ CFU g^−1^ were determined in two Genmaincha and Bacha weighable tea samples from Japan. 

Almost all the analysed black teas (*n* = 62) contained quantifiable amounts of mould. The mould counts reached at least 5.00 × 10^2^ CFU g^−1^ in sixteen samples, mostly in black teas from India and Sri Lanka (5.00 × 10^2^–9.60 × 10^3^ CFU g^−1^). Comparing the bagged (*n* = 30) and loose-leaf (*n* = 32) samples, the mean mould counts in the bagged tea samples (6.47 × 10^2^ CFU g^−1^) were higher than those in the loose-leaf tea samples (3.69 × 10^2^ CFU g^−1^). The weighable black tea samples mostly presented a lower range of maximum counts ranging between 1.00 × 10^1^ and 2.62 × 10^3^ CFU g^−1^.

The growth rates of the moulds were notably reduced in the oolong and Pu-erh tea samples, as indicated by notably lower mould quantities. These teas, especially the oolong variety, have shown antimutagenic efficiency against AFB_1_ toxins and toxins from other species [30,31]. The differences in their production methods may also explain their similarly lower rate of CFUs compared to nonfermented green tea. The mould counts in the seven tested oolong teas ranged between 1.00 × 10^1^ and 2.60 × 10^2^ CFU g^−1^, and between 1.00 × 10^1^ and 3.30 × 10^2^ CFU g^−1^ in the nine Pu-erh teas from the 20 total samples that tested positive for the presence of mould. The mean mould colony counts in the samples of these tea varieties varied around 6.85–7.55 × 10^1^ CFU g^−1^.

As presented in Figure 1, the mould counts of herbal teas exceeded the levels of fungal CFUs counted in *C. sinensis*, with the exception of linden tea (mean value: 7.20 × 10^2^ CFU g^−1^). The mean values of the mould unity contents counted in chamomile, peppermint, and hibiscus and dog-rose teas were around 1.29 × 10^4^ CFU g^−1^, 8.52 × 10^4^ CFU g^−1^, and 2.40 × 10^4^ CFU g^−1^, respectively.

### 2.3. Characterisation of Fungal Genera and Species

The fungal identifications were determined using characteristic identification keys reported in the literature [32,33]. Confirmation was provided using MALDI-TOF-MS equipped with the Bruker Daltonik library (2015), as described in Section 5.3.2 of this paper.

The study results indicated 17 fungal genera and 32 individual fungus species observed in the 138 mould-positive tea samples. Figure 2 summarises the eight most common fungal genera that were identified in the analysed teas. Five *Aspergillus* spp. and one *Penicillium* sp. were determined to be the most dominant fungal species and were identified in more than 10% of the analysed tea samples (Table 2). Some of *Aspergillus* spp., one *Fusarium* sp., and one *Lichthemia* sp. were found in only 5 to 7.5% of the teas, whereas most of the identified fungal species (*n* = 22) were found in one to two samples (less than 5% of the teas).

*A. niger* and *A. glaucus* isolates and *Penicillium sp.* (respectively, *P. chrysogenum*) were prevalent in most of the *C. sinensis* samples (with the exception of Pu-erh) and most of the herbal teas (Figure 3). These data are in good agreement with the results reported by other authors [12,34]. The predominance of *Aspergillus* species in *C. sinensis* tea products might be associated with their production during the fermentation process [35,36].

As presented in Figure 3, *A. niger* and *A. glaucus* were identified in 57 black tea varieties and 36 green tea varieties (*A. niger* in 29 samples and *A. glaucus* in 24 samples). In comparison, only three Pu-erh samples, three to seven oolong tea samples, and one to seven herbal tea varieties were positive for these two fungus species.

*A*. *niger* and *A. flavus* are the primary fungi of interest as sources that produce OTA and AFs [34,37]. However, not all isolates of mycotoxigenic fungi species have the genetic potential to produce mycotoxins. Storari et al. (2012) published a study of herbal teas available on the Swiss market. Within other *Aspergillus spp*. species, 41 isolates of *A. niger* were obtained during this study, whereas only 7% of them were able to produce OTA in teas [37].

In the current study, *A. flavus* was observed in fourteen black teas but in only two green teas and three herbal teas (one chamomile, one peppermint, and one linden tea). The fungi able to produce AFs were not found in the analysed Pu-erh and oolong tea samples.

The other characteristic fungal species determined in the black tea varieties included *P. chrysogenum* (*n* = 9), *Lichthemia corymbifera* (*n* = 8), *A. versicolor* (*n* = 7), and *A. nidulans* (*n* = 7). In other *C. sinensis* varieties, only *A. niger* and *A. glaucus* predominated.

The most typical genera in herbal teas, besides the two *Aspergillus* (*A. niger*. and *A. glaucus*) species, were *Mucor*, *Fusarium*, *Cladosporium,* and *Alternaria* (Figure 3).

*A. flavus* and other *Aspergillus.* spp. were found in only one to two samples of herbal teas (for example, *A. flavus* was determined in only one chamomile tea, one peppermint tea, and one linden tea sample, and was mainly found together with *A. versicolor* or *A. amstelodami* species).

Only one or two nonfilamentous (or *Aspergillus.* spp. and *Penicillium* spp. or *Rhizopus* sp.) fungi were observed in the hibiscus, dog-rose, and linden samples.

Chamomile and peppermint teas were determined to be the most contaminated with different fungal species. However, most of these fungal species were found in only one to three samples of these tea varieties.

*Alternaria* fungi were observed in four peppermint teas and one chamomile tea blend containing peppermint, rooibos, and other ingredients. In addition, *Phoma sorghina*, a common phytopathogen of plants, was identified in three pure peppermint teas and one tea composed of chamomile (34%), peppermint (33%), and fennel (33%). These peppermint teas were produced in Poland, and our results were in good agreement with the data provided by Zimowska (2007), who reported the typical prevalence of both *Alternaria* and *Phoma* species in cultivated peppermint plants [38]. There are insufficient data on potential human infections due to *Phoma* species, whereas the potential risks of *Alternaria* fungi related to the production of toxic contaminants are an emerging issue [39].

*P. chrysogenum*, *Lichthemia corymbifera*, and *Scopulariopsis brevicaulis* were common in three of the chamomile teas. Among the other fungi, only two fungi from the *Fusarium* genus (*F. incarnatum, F. equiseti*), identified in the four *C. sinensis* teas (Figure 2), were also found in the three peppermint samples and one chamomile tea sample.

While a few of the fungal species were common (e.g., *Aspergillus* spp.), this study also indicated the presence of other mycotoxigenic fungi, which raises the concern of potential mycotoxin production in the tea samples.

### 2.4. Mycotoxins’ Prevalence and Concentrations in Teas

The results using 2D-LC-TOF-MS indicated that 58% of the analysed teas (i.e., 44 black, 26 green, all oolong samples, 2 chamomile, 3 peppermint, 3 formulated hibiscus and dog-rose blends with fruit, and 3 linden tea samples) were mycotoxin-free.

Twenty mycotoxins were determined in 42% of the tea samples (*n* = 70). Almost a third of the tea samples were contaminated with one to three mycotoxins, while in most cases, the co-occurrence of more than four mycotoxins was observed. In total, 15.7% (*n* =11) of the 70 mycotoxin-positive tea samples were positive for more than 10 mycotoxins: one “gunpowder” green tea contained 13 mycotoxins, nine Pu-erh tea samples contained 10 to 16 mycotoxins, and one supermarket-brand peppermint tea contained 16 mycotoxins.

The following total incidences of the twenty determined mycotoxins were identified in the 70 mycotoxin-positive tea samples: DON (65%) and its derivates 15-AcDON (66%), 3-AcDON (54%), and D3G (37%), AFB_1_ (36%), AFB_2_ (27%), AFG_1_ (23%), AFG_2_ (10%), OTA (24%) and its dechlorinated derivate OTB (24%), STC (19%), ENN B (37%), ENN B_1_ (31%), ENN A (14%), ENN A_1_ (10%), ALT (24%) and ATX I (19%), ZEN (10%), T-2 toxin (3%), and HT-2 toxin (3%). Figure 4 summarises these data for the different tea varieties analysed in the present study.

As indicated in Figure 5, DON and its acetylated (3-AcDON and 15-AcDON) and glucuronidated (D3G) derivatives were found in concentrations at least one order of magnitude higher than the other mycotoxins, which were observed in the tea samples at average levels ranging from 1 to 100 µg kg^−1^ (Figure 5).

AFB_1_ (a Class 1 human carcinogen) and the other AFs (AFG_1_, AFB_2_, and AFG_2_) were determined in the following order in the analysed *C. sinensis* teas: Pu-erh tea (*n* = 8) > green tea (*n* = 7) > black tea (*n* = 6). The overall concentrations of AFB_1_ in these teas ranged between 0.22 and 39.0 µg kg^−1^ and between 1.37 and 102 µg kg^−1^ for the sum of the four AFs.

The green tea samples contained 0.46 to 2.99 µg kg^−1^ of AFB_1_. The four AFs co-occurred in five of the green tea samples, with the sum of their concentrations reaching up to 7.67 µg kg^−1^. One sample also contained AFB_1_, AFB_2_, and AFG_1_, with a total concentration of 1.54 µg kg^−1^. AFB_2_ and AFG_2_, which are less toxic, appeared at 1.5- to 3-fold lower concentrations compared to AFB_1_, while AFG_1_ presented similar concentrations in most of the samples.

The black tea samples of the current study also contained relatively low AFB_1_ concentrations from 0.39 to 3.45 µg kg^−1^. The sample with the maximum determined AFB_1_ contents also contained traces of AFB_2_ (0.28 µg kg^−1^). There were only two exceptions among the black tea samples, which both contained elevated concentrations of AFs. The first sample was a loose-leaf tea from Kenya, with 13.9 µg kg^−1^ of AFB_1_ and 12.1 µg kg^−1^ of AFB_2_. The second tea sample was an Indian spiced black tea blend (masala chai) that contained 39.0 µg kg^−1^ of AFB_1_, 11.1 µg kg^−1^ of AFB_2_, 41.7 µg kg^−1^ of AFG_1_, and 9.95 µg kg^−1^ of AFG_2_. AFB_1_ and AFG_1_ were the main toxins affecting the total contamination level (102 µg kg^−1^) of the sum of the four AFs in this tea sample. These levels may have been caused by the contamination of some spices (cardamom, cinnamon, ginger, etc.) used in the production of this tea mix.

Seven of the eight aflatoxin-positive Pu-erh tea samples contained relatively low levels of AFB_1_ (1.14–2.73 µg kg^−1^), AFB_2_ (0.27–0.79 µg kg^−1^), and AFG_1_ (0.99–1.63 µg kg^−1^). The sum of their concentrations ranged from 2.04 to 4.45 µg kg^−1^. AFG_2_ was not determined in any of the Pu-erh samples, similar to the black tea varieties. This result may be associated with the specifics of the fermentation conditions. One sample of Pu-erh tea (Yunnan Yiwu Spring Shen) presented higher levels of AFB_1_ (9.17 µg kg^−1^), AFB_2_ (2.76 µg kg^−1^), and AFG_1_ (7.9 µg kg^−1^).

AFs were observed in only four herbal tea samples. Hibiscus mono tea and the blend formulated with dog-rose and dried fruits contained low AFB_1_ concentrations of 0.22 and 0.4 µg kg^−1^. Relatively low levels of AFB_1_ at 0.98 µg kg^−1^ and AFG_1_ at 0.39 µg kg^−1^ were determined in the herbal blends of chamomile (27%) with rooibos (25%), peppermint (20%), and other ingredients. The multi-contaminated peppermint sample with 16 mycotoxins was positive for all four AFs at the following concentrations: AFB_1_: 5.62 µg kg^−1^, AFB_2_: 1.71 µg kg^−1^, AFG_1_: 6.02 µg kg^−1^, and AFG_2_: 2.36 µg kg^−1^.

Sterigmatocystin (STC), a possible human carcinogen (Class 2B toxin), is a common precursor of AFB_1_. STC co-occurred at low levels (1.2–1.8 µg kg^−1^) with the AFs in three of the green tea samples and at 13.0 ug kg^−1^ in the mycotoxin-rich peppermint tea sample, which contained four AFs. STC also co-occurred at trace to moderate levels from 1.06 to 93.4 µg kg^−1^ in nine of the Pu-erh samples, whereas the highest concentration was found in the Pu-erh sample that was also contaminated with elevated AF concentrations. These results for different tea matrixes confirmed the potential coincidence between STC and AFs due to the similar parental fungi (*Aspergillus* spp.) that produce these mycotoxins.

OTA, another Class 2B carcinogen, has also been commonly observed in teas, herbs, and their infusions due to its potential formation under improper storage conditions.

Only 9% (*n* = 4) of the green tea samples were positive for OTA at concentration levels between 1.1 and 3.4 µg kg^−1^. The co-occurrence of OTA with a less-toxic dechlorinated metabolite (OTB) was observed at concentrations ranging between 0.6 and 2.5 µg kg^−1^ in three of the green tea blends with fruit and natural flavourings.

In total, 6% (*n* =4) of the black tea samples were also positive for OTA (1.7–7.7 µg kg^−1^), whereas OTB was not observed in the black tea varieties.

However, the Pu-erh samples were positive for both OTA and OTB. Five of the Pu-erh samples contained low OTA levels ranging between 0.9 and 4.1 µg kg^−1^. OTB was determined in twelve Pu-erh teas at 0.8 to 30 µg kg^−1^. Three of these samples presented the co-occurrence of OTA and 25 to 30 µg kg^−1^ of OTB.

Comparing the data for the herbal teas, OTA was determined only in two chamomile blends at concentrations of 1.8 to 2.9 µg kg^−1^ and in two peppermint teas at 3.7–4.2 µg kg^−1^. These samples also contained 3.5 to 4.6 µg kg^−1^ of OTB.

DON, T-2, and HT-2 toxins, as well as ZEN, belong to the group of Fusarium mycotoxins that is regulated in agricultural products based on cereals. Their production is mainly associated with plant illnesses, and the environmental factors affecting plant microflora and accidental contamination may induce their production in tea plants, which has rarely been studied [13,14,15,16,40]. DON derivatives (3-AcDON, 15-AcDON, and D3G) are also commonly found in cereals. One of the concerns in terms of possible human health issues is the high water-solubility of DON and the potential extraction of DON derivatives into tea beverages [2,14,15,16].

Pu-erh teas showed the highest prevalence of Fusarium and mycotoxins compared to the other tea samples analysed within the present study (Figure 5). The D3G, 3-AcDON, and 15-AcDON derivatives co-occurred with DON in all twenty Pu-erh tea samples. The total concentration of the sum of DON and its three modified forms (D3G, 3-AcDON, 15-AcDON) (hereafter, ∑(DON derivatives)) in the Pu-erh samples ranged between 581 and 17,360 µg kg^−1^. The individual concentrations of DON, D3G, 3-AcDON, and 15-AcDON ranged from the lowest levels of 145, 113, 145, and 122 µg kg^−1^ to the highest contents of 8946, 3601, 4197, and 2252 µg kg^−1^. Except for a few exceptions featuring predominant DON contents, these toxins presented similar concentration levels in most of the Pu-erh samples.

Comparing the varieties, two were raw Pu-erh teas, which are commonly fermented under hot and humid surroundings, thereby leading to the formation of fungi: these two samples showed DON concentrations 2 to 4 times higher (6255–8946 µg kg^−1^) than those determined in the ripened Pu-erh tea samples of the present study. The DON concentrations in the ripened Pu-erh samples ranged between 145 and 5808 µg kg^−1^.

Fifty-nine percent (*n* = 10) of the green tea samples, mostly the organic green leaf teas from China and Japan, were positive for DON and one to three of the DON derivatives. DON was observed at concentrations ranging from 97.1 to 859 µg kg^−1^—up to 10 times lower compared to the maximum concentrations determined for the Pu-erh tea samples. Five of the green tea samples also contained all of the four DON mycotoxins (their combined concentrations ranged between 1446 and 4512 µg kg^−1^), and three of the samples contained both of the acetylated DON derivatives, 3-AcDON and 15-AcDON, with the combined concentrations ranging from 404 to 3086 µg kg^−1^. Two of the green samples contained only individual acetylated DON derivatives at relatively low concentrations of 61.7 µg kg^−1^ (3-AcDON) and 211 µg kg^−1^ (15-AcDON).

3-AcDON was the predominant mycotoxin in the analysed green tea samples at concentrations of 213–1792 µg kg^−1^, which was 1.6 to 4.9 times higher than the levels of DON in the analysed tea samples. For example, the concentrations of 15-AcDON ranged between 58.5 and 1248 µg kg^−1^ and were mostly at the same level as the DON contents in each of the tea samples, with two exceptions. The exceptions included an organic sencha tea, with concentrations of D3G, 15-AcDON, and 3-AcDON exceeding the concentration of DON (126 µg kg^−1^) by 2.7, 2.8, and 4.9 times, respectively. The second exception was also a sencha tea mixed with different nuts and sugar-coated pineapple; in this case, the concentrations of D3G, 15-AcDON, and 3-AcDON derivatives exceeded the concentrations of DON (464 µg kg^−1^) in the tea blend sample by 2.1, 2.7, and 3.9 times, respectively.

As indicated in Figure 4, the traditional black tea varieties were less contaminated with DON and the two acetylated DON derivatives. D3G was not found in any of the analysed black tea samples, possibly due to the lower stability of this compound under the thermal conditions of black tea production, and taking into account its lower stability compared to DON, which was noted in previous studies on DON derivatives in baked cereal products [41].

DON was determined in only 39% (*n* = 7) of the mycotoxin-positive black tea samples at concentrations from 26.5 to 211 µg kg^−1^. The acetylated derivatives showed similar levels of contamination to those of DON: 15-AcDON was observed in nine of the analysed black tea samples (with concentrations ranging between 25.8 and 189 µg kg^−1^), whereas 3-AcDON was determined in only three samples of this tea variety (with concentrations ranging between 106 and 428 µg kg^−1^). Comparing the co-occurrence of DON and its derivatives, only five of the black teas were positive for the co-occurrence of DON and 15-AcDON at summary concentrations from 116 to 290 µg kg^−1^, and two of the samples were positive for the co-occurrence of 15-AcDON and 3-AcDON at 290 and 537 µg kg^−1^. In comparison to green teas, 15-AcDON exceeded the levels of DON in only two of the black tea samples, as the derivatives were mostly present at lower or similar concentrations to those of DON.

DON and its derivatives were also observed in 60% (*n*= 9) of the 15 total mycotoxin-positive herbal teas, whereas the contamination levels in most cases were lower than those of the *C. sinensis* tea samples.

A blend of hibiscus with dried apple and liquorice was positive for 15-AcDON at 200 µg kg^−1^, whereas no other DON mycotoxins were determined in the hibiscus or dog-rose teas.

Six of the chamomile tea samples were positive for DON concentrations ranging between 44.0 and 166 µg kg^−1^. Three of the samples contained 3-AcDON (concentrations ranged from 30.6 to 146 µg kg^−1^), and four also contained 15-AcDON (with concentrations ranging between 30.6 and 146 µg kg^−1^). D3G was not found in any of the chamomile tea samples, whereas the combined concentrations of DON and DON acetylated derivatives ranged between 81.1 and 718 µg kg^−1^, respectively.

Three of the peppermint teas were characterised as being the most contaminated with DON and its derivatives compared to other herbal species. Two of the samples contained three DON mycotoxins, with DON ranging between 52.56 and 521.68 µg kg^−1^, 3-AcDON ranging between 300 and 2227 µg kg^−1^, and 15-AcDON ranging between 96.5 and 1140 µg kg^−1^. The third peppermint sample was noted above for its high contamination prevalence of sixteen mycotoxins, which, besides the four AFs, also included all four DON mycotoxins, with the concentrations of DON, 3-AcDON, 15-AcDON, and D3G reaching 953, 2255, 2227, and 740 µg kg^−1^, respectively (the combined concentration of all four DON compounds reached 5631 µg kg^−1^ in this peppermint tea sample).

ZEN, a common estrogenic toxin, was not found in most of the analysed tea samples, except for the seven Pu-erh samples, which contained 4.72 to 56.1 µg kg^−1^ of ZEN.

T-2 and HT-2 toxins were identified in only two green tea samples, with 5.58 µg kg^−1^ of HT-2 and 8.1 µg kg^−1^ of T-2, and in one peppermint tea sample, which contained both T-2 and HT-2 toxins at 4.90 and 37.5 µg kg^−1^ concentrations, respectively.

Enniatins (ENNs) and beauvericin (BEA) are emerging *Fusarium* mycotoxins frequently found in grain products; these mycotoxins raise potential concerns due to their chronic toxicity upon long-term exposure [42,43]. BEA was not observed in any of the analysed 166 tea samples. Further, the black and oolong tea samples were free of ENNs, whereas the other samples, such as the Pu-erh, green, and herbal teas, indicated the co-occurrence of different ENNs.

ENN B was the most predominant among this class of toxins, with levels of 0.93 to 143 µg kg^−1^ in 15.7% of the ENN-positive samples (*n* =26) from the 166 analysed teas. ENN B_1_ was the second most frequently occurring ENN and was present in concentrations between 0.5 and 201 µg kg^−1^ in twenty-two tea samples.

ENN A_1_ and ENN A were found in only five and ten samples of Pu-erh teas, respectively (ENN A ranging between 0.69 and 115 µg kg^−1^). However, 4.33 µg kg^−1^ of ENN A_1_ was also observed in one of the green tea samples and at a level of 55.3 µg kg^−1^ in the multi-contaminated peppermint tea. 

ENN B ranged between 0.93 and 18.7 µg kg^−1^ (the mean value: 5.96 µg kg^−1^) in the positive green teas (*n* = 7), while five of the samples were also positive for the co-occurrence of ENN B_1_ ranging between 6.34 and 47.5 µg kg^−1^ (mean value was 18.8 µg kg^−1^).

A sheng Pu-erh sample with exceptionally elevated ENN and ENN B_1_ contents (143 and 125 µg kg^−1^) was also observed, whereas in the other ten ENN-positive Pu-erh samples, the concentrations of both toxins were similar to those found for green tea varieties. Concentrations of ENN B ranging between 1.28 and 31.4 µg kg^−1^ (mean value: 10.1 µg kg^−1^) were determined in these Pu-erh tea samples, and ENN B_1_ concentrations between 0.62 and 22.6 µg kg^−1^ (mean value: 7.97 µg kg^−1^) were observed.

Three chamomile teas, one peppermint tea, and two dog-rose blends with hibiscus and fruits presented low concentrations of ENN B and ENN B_1_ ranging between 0.98 and 11.3 µg kg^−1^. Both ENNs were present at notably higher levels in the two peppermint teas. One was a tea sample from Latvia, which contained 28.3 µg kg^−1^ of ENN B and 111 µg kg^−1^ of ENN B_1_. The multi-contaminated peppermint tea contained 83.9 µg kg^−1^ of ENN B and 201 µg kg^−1^ of ENN B_1_. 

Here, the presence of two emerging Alternaria mycotoxins (ALT and ATX I) was observed for the first time in commercial tea samples available on the Latvian market. Fourteen of the analysed Pu-erh tea samples were positive for both ALT and ATX I at concentrations of 0.10–9.10 and 1.44–28.3 µg kg^−1^. In the six samples, both mycotoxins co-occurred. ALT and ATX I were both observed in only two black teas (ALT ranging between 0.8 and 1.3 µg kg^−1^) and three green teas (ALT ranging between 1.1 and 3.7 µg kg^−1^; one of the samples also contained ATX I at 2.72 µg kg^−1^).

Among the herbal teas, only one formulation of chamomile with rooibos, peppermint, and other ingredients was positive for ALT at 3.9 µg kg^−1^. Moreover, one peppermint tea contained 4.81 µg kg^−1^ of ATX I, while the multi-contaminated peppermint sample was positive for 7.10 µg kg^−1^ of ALT and 5.78 µg kg^−1^ of ATX I.

### 2.5. Dietary Exposure Studies of Mycotoxins

A dietary exposure assessment was conducted according to the procedure described in Section 5.5. Of course, the extraction rate of mycotoxins from dry matter into beverages during infusion should be considered to evaluate exposure risks [2]. As only dry tea samples were analysed in the present study, the worst-case scenario that the mycotoxin concentrations found in the dry samples would be fully extracted into the infusions was assumed. The upper-bound values and the maximum concentration levels found in the analysed tea varieties were included in the evaluation. This approach was based on our previous studies, which indicated a high extraction rate (30 to 100%) of DON and ZEN from dry products into tea infusions [16]. Our previous study indicated 32% extraction of DON (41.4 µg L^−1^) from a dry dog-rose sample into its infusion, whereas a 100% extraction of up to 273 µg L^−1^ of DON was observed for several herbal teas. These results are in good agreement with the data on DON’s water solubility (55 mg mL^−1^) reported in a comprehensive review [2]. Pallarés et al. (2017) reported on 15-AcDON (found at 60.5–61 µg L^−1^) in tea beverages [14]. This report raised concerns of the potential co-extraction of DON and its derivatives found in the tea samples of the present study, taking into account the risk assessment studies published by the European Food Safety Authority (EFSA) that found the same tolerable daily intake level (TDI) to be 1 µg kg^−1^ bw day^−1^ for both DON and the sum of DON and its derivatives (D3G, 3-AcDON, and 15-AcDON) [44]. The results of the screening studies were used for the selection of potential compounds with high prevalence and contamination levels. Most of the mycotoxins occurred in the teas at concentrations too low to have potential risks upon acute intake.

DON and its derivatives were found in green, black, and Pu-erh teas (especially the elevated concentrations of these mycotoxins in the multi-contaminated peppermint tea with the 16 co-occurring mycotoxins); ZEN, observed only in the Pu-erh tea samples, and OTA, found in most of the teas, were included in the dietary intake assessment, due to their relatively high prevalence in the commercial teas analysed in this study (Table 3).

As noted in the description of potential acute daily intake calculations (Section 5), the average consumption rates of nonfermented (green tea), fermented (black tea), and herbal (peppermint) teas were obtained from the data from our recent study [18].

The approximate average volume (200 mL) and mean dry tea amount (2.00 g) per serving of a hot water infusion (allowing the tea to infuse for an average of 5 min) were used (i.e., the usual procedure recommended for most of the analysed teas) [2]. It should be noted that this assessment applies to daily tea drinkers, without considering the use of teas for medical purposes. The lack of information on the consumption of Pu-erh teas required additional approximations; we assumed that the preparation method for infusion and consumption is the same for the closest fermented tea variety (black tea). Although the Pu-erh tea brewing procedure might be different, in this assessment, similar conditions were evaluated within the worst-case scenario (Table 3).

The probable daily intake (PDI) values were calculated from the average daily intake values of the Latvian population [18]. The average data for adults and the elderly (both male and female) were evaluated. The PDIs calculated within the approximated model conditions for the upper-bound values and maximum concentrations of evaluated mycotoxins are summarised in Table 3. While the data for the population indicated similar levels of black and green tea intake (318 and 345 mL), slightly lower levels were found for peppermint tea infusions.

The PDI values for acute intake of the maximum mycotoxin concentrations were 1.4 to 16.5 times higher than the calculated values for the mean concentrations as the potential contamination levels of infusions.

Considering the low concentrations of OTA found and the similarity of the data to the mean upper-bound and maximum levels, the PDI values reached only 0.17 to 2.05% of the OTA TDI, thus indicating little exposure potential during tea intake.

The values of AFB_1_ and summary AF PDI values were also calculated considering their high prevalence in 30% of the analysed teas. The levels were similar to those reported by other authors [45]. Due to their genotoxicity and carcinogenic effects, AFs have no TDI values, thus another concept based on the available benchmark dose lower confidence limit for a 10% increase in cancer incidence obtained from animal study data modelling was used. Comparing the PDI values with BMDL indicated that AF contents were safe for the consumers.

In the case of traditional black and green teas, the PDI values of DON and the summary contents of DON and DON derivatives were only 5.33 to 7.40 times higher in the case of maximum concentrations considered as contamination levels of beverages compared to that of upper-bound levels determined in tea varieties.

The PDI values of DON for traditional teas generally reached only 0.96 to 4.23% of the TDI, similarly to herbal teas, where the PDIs for the upper-bound level and maximum determined mycotoxin contents reached only 2.03 to 3.42% of the TDI.

The PDI values of the total DON and DON derivatives calculated for traditional teas and peppermint teas were 2.60 to 5.99 times higher than the PDI calculated for DON, which indicates the strong influence of multi-contamination on the overall DON intake. This was notably observed in Pu-erh teas, where the PDI values for the total contamination of DON and DON derivatives reached 27.4% and 78.9% of the TDI. The high rate of DON mycotoxin contents in Pu-erh teas should be considered, as amounts greater than 2 g (8 g approximately) are commonly used during the brewing process, and the first infusion is commonly not used.

In this study, the PDI values calculated for the approximate worst-case scenarios remained below the TDI, which indicates a low probability of intoxication via the acute intake of traditional commercial teas and peppermint beverages.

## 3. Discussion

The present study provided an assessment of the microbial quality and mycotoxin prevalence in relatively large datasets of traditional and popular *C. sinensis* and herbal tea varieties that were purchased from Latvian tea stores and supermarkets. The results were compared with recent data from other studies, providing new data on regulated and emerging mycotoxin contamination in teas related to the fungal species explored in the present study.

The microbiological data were in good agreement with recently reported results. Carraturo et al. (2018) reported on sixteen black and green teas: the mean total mould and yeast counts ranged between 8.00 × 10^0^ and 4.49× 10^3^ CFU g^−1^ in green tea varieties and 4.30 × 10^1^ and 3.06 × 10^3^ CFU g^−1^ in black tea varieties [12]. The higher contents of fungi in some teas, especially in some green tea varieties, might be related to the lower temperatures used during the production of nonfermented *C. sinensis* tea products, although there are several different potential factors [2].

The microbial quality found in the present study was in agreement with the safety recommendations for the total fungal counts (≤1.00 × 10^5^ CFU g^−1^) and yeast counts (≤1.00 × 10^6^ CFU g^−1^ and ≤1.00 × 10^5^ CFU g^−1^) set for both dry *C. sinensis* teas and dry herbal teas in the recommendations by the Tea and Herbal Infusions Europe (THIE) and the American Herbal Product (AHPA) associations [46]. Many countries have much stricter requirements and/or recommendations for the management of tea’s microbiological quality and safety for consumers.

The Regulation of the Cabinet of Ministers of the Republic of Latvia, No. 461 outlined limitations of ≤5.00 × 10^2^ CFU g^−1^ for the sum of the total number of moulds and yeast cells in dry *C. sinensis* teas [29]. Against these limitations, the mould counts in 28% of the traditional teas (i.e., the sixteen black tea samples and fourteen green teas from supermarket brands) exceeded this quantity by up to 19.2 and 31.6 times. However, this document does not regulate herbal teas and only provides a recommendatory function.

The Technical Regulation CU TR 021/2011, which is used in the Russian Federation, permits fixed mould counts of ≤1.00 × 10^3^ CFU g^−1^ for both *C. sinensis* and herbal teas, including medical plants (dry), and ≤5.00 × 10^1^ CFU g^−1^ mould counts for herbal teas consumed by pregnant and breast feeding woman or by children [47]. Several of the commercial teas included in the current study were imported from the Russian Federation and can also be found in Customs Union markets; thus, the microbiological results of the mould contents were compared to the aforementioned limitations: 17.6% (*n* = 28) of the tea samples (nine green, six black, five chamomile, five peppermint, two hibiscus, and one linden) exceeded the limitations by up to 90 times. The total mould counts in these teas ranged between 1.01 × 10^3^ and 9.00 × 10^4^ CFU g^−1^.

Our data for black tea and herbal samples were comparable with the recent results reported by Minaeva et al. (2019) [47]. In their study, the microbiological contamination of 54 raw *C. sinensis* and herbal teas (30 packed *C. sinensis*: teas, 6 weighable *C. sinensis* teas, and 18 weighable herbal teas) from markets in the Russian Federation was evaluated. This study indicated that only 16.7% of commercial black tea samples from India and Indonesia, and one sample from Sri Lanka were not in agreement with the microbiological limitations. These samples contained mould fungi concentrations up to 1.3 × 10^3^–8.2 × 10^3^ CFU g^−1^, whereas the levels in tea infusions dropped to nondetectable levels, indicating reduced threats to consumer safety. The authors of this study also reported that 10 out of 18 herbal teas had fungal mould contents ranging between 5.00 × 10^3^ and 1.00 × 10^6^ CFU g^−1^, which notably exceeded the standard limitations for total fungi, whereas the mould levels dropped to the relevant limitations after tea brewing, unlike the bacterial counts, which were not reduced in herbal teas after preparing the tea infusions [47]. The authors of this study also reported on the prevalence of *A. niger* in *C. sinensis* teas, noting the multi-contamination of more than five species, including *Aspergillus* sp., *Penicillium* sp., *Alternaria* sp., *Fusarium* sp., and *Cladosporium* sp., which were found in all herbal tea samples collected from the markets.

Compared to traditional black and green teas, the oolong and Pu-erh samples showed similarly low mould contents. No data were found from studies on the microbial assessment of dry oolong teas. However, some studies focused on cold green tea and oolong tea infusions have demonstrated the formation of *Cladosporium sp*. (*C. cladosporioides*) and *A. niger* species in these beverages [48]. Meanwhile, the microbial qualities of raw and ripe Pu-erh varieties have been frequently studied [6,49,50,51]. In the present study, most of the tested teas were ripened Pu-erh varieties and presented no moulds, although eight ripened Pu-erh samples contained very low mould counts (1.00 × 10^1^–1.80 × 10^2^ CFU g^−1^), and one raw Pu-erh sample contained a 3.00 × 10^2^ CFU g^−1^ mould count. Lower levels of mould were observed than those reported by other studies. *A. glaucus* and *A. niger* were the predominant species in six of the mould-positive Pu-erh samples, whereas one of the ripened Pu-erh teas was positive for *Cladosporium herbarum*, *Cladosporium chrysogenum*, and *Lichthemia corymbifera* species, which may be due to the aging conditions, as *Cladosporium spp*. are recognised to be slow-growing fungi [48].

The production of ripe Pu-erh tea includes an additional wet-piling fermentation process, which involves piling and aging stages that lead to the production of mould, bacteria, and fungi species, such as *A. niger* and *Blastobotrys adeninivorans,* ultimately yielding a specific stringent taste [49,50]. Zhao et al. (2010) reported much higher fungal colonisation levels found in ripe Pu-erh tea samples (1.00 × 10^4^–4.90 × 10^3^ CFU g^−1^) than in raw loose tea samples (1.6 × 10^3^–4.8 × 10^3^ CFU g^−1^), due to the fungal production during the wet-piling period [50]. 

Haas et al. (2013) reported a relatively high variety of fungi found in raw Pu-erh teas (1.0 × 10^1^–2.6 × 10^6^ CFU g^−1^); the results of the present study coincided with these results [51]. While the levels of fungi were mostly low, this study observed the presence of major mycotoxigenic fungi species, such as *A. niger*, *A*. *flavus,* and *Alternaria alternata*, which may affect the production of AFs, OTA, and *Alternaria* toxins.

The results for the AFB_1_ contamination found in the mycotoxin-positive samples were comparable to the screening results reported in a study on green teas from Moroccan markets, where concentrations from 1.8 to 41.8 µg kg^−1^ were found [11]. This study indicated a higher prevalence of AFB_2_ and other AFs, ranging from 1.20 to 75.41 µg kg^−1^ in these Moroccan teas [11] compared to the results of the current study, where AFs B_2_, G_1_, and G_2_ were present in only a few samples at low concentrations (≤2.55 µg kg^−1^). The low AF prevalence in green teas was in good agreement with the results of a study from Spain, which also indicated a low prevalence of AFB_1_ in commercial green teas [52]. Only one sample from among all ten analysed teas was positive for AFB_1_, with a concentration of 5.40 µg kg^−1^.

Considering their potential contamination during aging and wet-milling, Pu-erh teas are commonly evaluated for AF and OTA contamination. Haas et al. (2013) analysed the presence of AFs, fumonisins (FBs), and OTA in 36 Pu-erh samples, where AFs were not found, and only 11% OTA-positive samples were reported [51]. Dong et al. (2018) investigated 50 ripe and 50 raw Pu-erh samples, which were also negative for AFB_1_ [53].

The present study indicated a low prevalence of AFs in five of the Pu-erh samples. This result agrees with the study data reported by Li et al. (2015) on 14 raw and 14 ripe tea samples tested by HPLC-MS methods, which also indicated the absence of AFG_2_ in most of the ripe Pu-erh teas and 10 of the raw Pu-erh varieties [49]. AFB_1_ concentrations ranged between 0.345 and 8.333 µg kg^−1^ in the ripe Pu-erh teas and between 1.827 and 15.149 µg kg^−1^ in the raw Pu-erh varieties; these are comparable data to those reported in the present study. Similar data were also reported by other authors [54].

Notably, AFB_1_ and sum of four AFs are still not regulated in teas in Europe, although Moroccan regulations do define allowable quantities in teas, which coincide with the restrictions of 5 and 10 µg kg^−1^ of maximum concentrations set for these toxins in spices by European legislative authorities [11,55]. The specified restricted values for AFB_1_ in spices coincide with the regulatory limits for raw tea in the national regulations set by the Customs Union countries [2].

The European Pharmacopeia has set restrictions for the allowable AFB_1_ and combined four AF limitations in medical herbs, which are 2.5 times lower (2 μg kg^−1^ limit for AFB_1_ and 4 μg kg^−1^ for combined AFs) than the concentration limitations allowed by the EC for condiments [16,55]. Thus, the levels of AF and the sum of the four AFs exceeded these limitations 2.8- to 3.9-fold; these limitations also apply to teas, according to the regulations of the Customs Union [2]. These data coincide with our recent results, which also presented a low rate of contamination with AFs in medicinal herbal teas, with the only exceptions being detox teas based on senna and alder buckthorn, which notably exceeded the levels recommended by the European Pharmacopeia [16,56].

Comparing fungi and mycotoxin prevalence, only four of the tea samples (16%) showed the co-presence of A. flavus species and AFB_1_, whereas most of the teas indicated low contamination with AFs, with the exception of two samples with elevated AF contents (a black tea from Kenya and Indian masala chai), which both had concerning levels of AF exceeding the acceptable limitations set by the relevant worldwide regulations [2,16,55,56].

However, most of the AF-positive samples also contained detectable levels of carcinogenic STC, mostly at low levels that were in good agreement with other studies. Chaly et al. recently reported STC at trace levels (0.4 µg kg^−1^) in two samples of black tea [57]. Our recent study based on 1D-LC-TOF-MS also indicated the co-occurrence of AFs and STC at low levels (0.42–14.2 µg kg^−1^) in medicinal herbs [16].

Kiseleva et al. (2020) found STC levels below the LOD in one black tea sample and six herbal blends (concentrations ranging between <4 and 10.0 µg kg^−1^) collected from tea markets in the Russian Federation, whereas no traces of this mycotoxin were found in the green and Pu-erh teas included in this recent study [58]. The co-occurrence of both AFs and STC in the present study in mycotoxin-positive green teas (*n* =3), Pu-erh teas (*n* = 3), and mycotoxin-rich peppermint tea raises concerns over their possible cumulative effects, which should be taken into account when evaluating mycotoxins’ impacts on human health. It is well known that several *Aspergillus* species, including the species indicated in the present study (e.g., *A. flavus*, *A. nidulans*, *A. versicolor*, etc.), can produce STC [59]. While the presence of *A. glaucus* and *A. versicolor* was identified in five of the thirteen STC-positive tea samples, these data should be further evaluated using more advanced methods of fungal genus analysis before conclusions can be drawn.

OTA raised the most notable concern due to the high prevalence of *A. niger* species in the analysed tea samples, especially in traditional *C. sinensis* varieties (e.g., black tea: 98.4%). Additional studies based on PCR and molecular tools will be necessary to genetically determine the isolates able to produce mycotoxins, as reported by a recent study [37]. The evaluation of data indicated that eight (53%) of the fifteen OTA-positive samples had a prevalence of both OTA and *A. niger*. However, only eight cases from a total of 105 tea samples were contaminated with this fungus. Indeed, these data coincided with the low case rates of OTA-producing *A. niger* isolates found in Reference [37].

It can be concluded that the contamination rate of OTA in the mycotoxin-positive tea samples in this study was comparable to that reported by other researchers, though it must be noted that OTB was not reported in teas in any previous studies. Thus, only the results of OTA content can be compared.

The distribution of OTA reported by Haas et al. (2013) ranged between 0.65–94.7 µg kg^−1^ in 4 (11%) of the 36 Pu-erh tea samples analysed [51].

Carraturo et al. (2018) studied the OTA levels in dry leaves of black and green tea and their beverages, with the levels in both dry tea varieties ranging between 0.01 and 21.49 µg kg^−1^ in 69% (*n* =22) of the analysed tea samples [12]. This study also observed fungal contamination by *A. niger*, with total mould counts of 1.00 × 10^2^ to 2.8 × 10^5^ CFU g^−1^ in tea. This study indicated a higher transfer rate (54.47 ± 14.48%) for the green tea compared to the OTA in black tea infusions (33.65 ± 4.37%) [12].

The current results for OTA in herbal teas also coincided with our recent report of OTA levels ranging between 2.86 and 30.3 µg kg^−1^ in medicinal herbs collected from Latvian drug stores [16]. However, OTA was not found in any of the tea beverages analysed in this study, a result which was similar to that in the report on herbal and *C. sinensis* infusions by Pallarés et al. (2017, 2019) [14,15].

The results for the OTA content in peppermint teas were comparable to those reported by Santos et al. (2009), whose study also indicated a low contamination rate of OTA in chamomile and spearmint. The OTA levels ranged from 0.1 to 1.4 µg kg^−1^, and the maximum concentration in herbs from Spain reached only 17.3 µg kg^−1^ [40].

In Europe and other countries, OTA is not regulated in teas or herbs. The only available regulations for the maximum OTA levels in plant matrixes include limitations of 15 µg kg^−1^ in paprika and other spices, and 5 and 10 µg kg^−1^ in ground roasted and instant coffee, respectively [60]. Notably, the overall levels of 0.9 to 7.7 µg kg^−1^ were too low to raise concerns from this perspective.

A recent study also noted that coffee samples from a Latvian market did not exceed the maximum permitted levels of OTA in the European Union (5.0 µg kg^−1^) [61].

Malir et al. (2014) reported the results of a survey of OTA in black and herbal teas (*n* = 12 each) and ground roasted coffee naturally contaminated at 0.92 µg kg^−1^ [8]. Their study indicated OTA levels up to 250 µg kg^−1^ in dry black teas and up to 104 µg kg^−1^ in dry herbal teas; these are notably higher levels than those reported in the present study. However, the analysis of tea infusions prepared from the samples with the highest OTA contents indicated a low transfer rate of mycotoxins into infusions—34.8 ± 1.3% from black tea and 4.1 ± 0.2% from fruit tea [8]. OTA transfer into coffee also did not exceed 66.1%, which indicates that in all cases, the contamination rates did not pose substantial human health risks.

There is good agreement between the conclusions of previous acute dietary exposure studies and the present study, as the PDI levels calculated for the worst-case scenario, even at the maximum OTA levels, remained below 2.06% of the TDI for OTA, in line with EFSA recommendations.

While only a few recent studies have included Fusarium and Alternaria mycotoxins in their surveys for tea contamination, 1D-LC-MS mostly indicated a low sensitivity and issues related to the proper separation of 3-AcDON and 15-AcDON [58], as well as the need for multistep sample preparation to overcome these issues [15]. The developed 2D-LC method, which was applied within the present study, notably improved the separation efficiency of different DON metabolites and other challenging mycotoxins [17]. This was also indicated when comparing the efficiency of mycotoxin quantification in this study compared to the efficiency in previously reported studies on mycotoxins in herbal teas [16].

The results for DON and its metabolites in herbal teas were comparable with the data reported in a previous study [16] that used 1D-LC-TOF-MS to analyse the mycotoxins in medicinal herbs, in that more than half of the teas were contaminated with DON at concentrations ranging between 129 and 5463 µg kg^−1^ [16]. Santos et al. (2009) also observed DON in chamomile tea at concentrations from 35.2 to 242 μg kg^−1^, which agrees with the results of the present study, although this method also did not include an analysis of DON derivatives [40].

The same results were found for T-2 and HT-2 levels, which coincided with those reported in Reference [16], whereas the concentrations of those toxins did not exceed 60.3 µg kg^−1^ in medicinal herbs.

Santos et al. (2009) reported T-2 contamination in 88% of the herbal species analysed, with concentrations ranging between 0.28 and 260 µg kg^−1^ [40].

Kiseleva et al. (2020) also reported different Fusarium fungal metabolites among the 29 total mycotoxins tested in 24 *C. sinensis* teas and 24 herbal teas [58]. This study reported two firefly tea samples that were positive for DON levels below the LOQ (<1250 µg kg^−1^), while no traces of DON were found in any other teas tested within this study. While AcDON metabolites cannot be separated using the method reported in this study, two packaged samples of green tea and three samples of black tea were identified to be positive for some of the acetylated DON metabolites found below the LOQ (<250 µg kg^−1^) [58]. The results of the study by Kiseleva et al. (2020) are in good agreement with the results of the present study and also indicated the low prevalence of T-2 toxins in tea samples. Here, only three samples were positive for T-2 below the detection level, whereas Kiseleva et al. identified a metabolite of a T-2 triol that was found in one herbal blend sample at 10.00 µg kg^−1^ [58].

Pallarés et al. (2017, 2019) reported that neither T-2 nor HT-2 could be found in *C. sinensis* and herbal teas or their infusions, whereas 15-AcDON was found at 60.5–61.0 µg L^−1^ in two green tea blends with peppermint [14] and at 112.5 µg L^−1^ in one valerian tea sample [15]. The acute dietary exposure studies indicated that the PDIs were far from the TDI.

The dietary risk assessment provided in the current study also indicated a low rate of potential PDI for most traditional and peppermint teas, which agreed with the previous data reported in Reference [16] and those of previous infusion studies [14,15]. Attention should be given to the multi-occurrence of DON metabolites at relatively high concentrations, especially in Pu-erh samples.

Multi-mycotoxin surveys on Pu-erh teas did not indicate DON or DON derivatives, while the current data on the low ZEN contents in Pu-erh tea were in good agreement with the recently reported data in Reference [6]. Considering the low contamination rates, the PDI values found for Pu-erh teas were far from the TDI found for acute exposure to ZEN.

The emerging Fusarium mycotoxins (four ENNs) and Alternaria mycotoxins have been frequently assessed in different agricultural plant species.

ENNs presented the second highest frequency of co-occurrence after DON and DON metabolites: at least two ENNs were found together in 85% (*n* = 22) of the 26 teas contaminated with ENNs, mostly in Pu-erh and herbal teas. ENNs were observed together with DON and DON derivatives in 85% (*n* = 22) of the samples. The co-occurrence of different *Fusarium* mycotoxins has been commonly described for more frequently evaluated matrixes of grains and cereal products [62].

While the present study indicated no traces of BEA in the 70 mycotoxin-positive samples, Kiseleva et al. (2020) reported a relatively high co-occurrence of BEA with ENN A and ENN B, mainly in herbal teas [58]. This survey of 29 mycotoxins in tea products from the Russian Federation reported that BEA was found in 54% (*n* =13) of the 24 *C. sinensis* tea samples at concentrations of <2.5–6.00 µg kg^−1^. Within this investigation, ENN B was the predominant ENN toxin, which was found in single-herbal teas at concentration levels between <2.5 and 26 µg kg^−1^, and in six of the herbal blends at concentrations from 13.6 to 55.0 µg kg^−1^ [58].

These results agree with the contamination levels of ENN B found in the present study and are also in good agreement with our earlier study of medicinal herbs, which indicated a high prevalence of ENN B in 55% of 60 analysed teas, where the concentrations mostly were below 11 µg kg^−1^ [16].

While we did not find any reports of ENNs in green and Pu-erh teas, a study by Pallarés et al. (2017) indicated one case of a green tea beverage with ENN B levels below the LOQ [14]. A subsequent study from the same authors also reported ENN B to be the only ENN found in horsetail infusions, which was also present in the samples below the quantification level [15].

The present study provided data on the presence of *Alternaria* toxins in *C. sinensis* and herbal teas. As discussed in the mycotoxin analysis, ALT and ATX I toxins were mainly found in the Pu-erh tea and green tea samples, and in only a few herbal (mainly peppermint) teas. *Alternaria alternata* species were observed in only five of the chamomile and peppermint teas, whereas in three samples (in one chamomile and two peppermint tea samples), both fungi and their metabolites were found.

In a recent review, Crudo et al. (2019) suggested that *Alternaria* toxins commonly co-occur with AFs and emerging ENNs in different condiments [39]. Co-occurrence with AFs was found in 73% (*n* = 16) of the samples of the 22 teas with both observed *Alternaria* toxins. These samples mainly included Pu-erh varieties (*n*= 10), green and black teas (*n* =3 and *n* = 2), one chamomile blend, and the peppermint tea sample contaminated with 16 co-occurring mycotoxins.

No other studies reporting on *Alternaria* toxins in tea samples were found, except for the interesting results reported in Reference [58]. Besides ENNs and the other reported mycotoxins, the present study also included three *Alternaria* toxins (TE, AME, and AOH), with the LOQ values for their quantification reaching 8, 750, and 1000 µg kg^−1^ [58]. This study reported traces below the LOQ for the AME and TE found in one green tea, one black tea, two thyme teas, and two peppermint teas, which indicated the co-occurrence of both AME and TE [58]. These data indicate the need to further study these emerging mycotoxins and regulated toxins due to their potential cumulative toxic effects. This study’s data agree with the conclusions drawn in Reference [58] that a low prevalence of fungal species and low mould counts does not guarantee the absence of mycotoxins, and that co-occurring groups of mycotoxins should be evaluated in further studies, taking into account the high prevalence of both regulated and nonregulated toxins in commercial tea samples.

## 4. Conclusions

The current food safety regulations for the European market provide few mandates related to foodborne and environmentally produced toxins in traditional teas and herbal teas used daily. Although there are some regulations available for AFs, multi-contamination should be considered. This study demonstrated advances in the 2D-LC-TOF-MS method for the simultaneous quantification of regulated and emerging mycotoxins produced by *Alternaria*, *Aspergillus*, *Penicillium,* and *Fusarium* strains and their secondary derivatives in 166 commercial teas, including traditional and novel Pu-erh and oolong teas, as well as common herbal teas (pure chamomile, peppermint, linden, hibiscus, and dog-rose teas, as well as formulated blends) available at retail stores and from tea houses. The microbial quality studies indicated that most of the teas were in good agreement with the recommendations of European institutions and with Latvian regulations of tea product quality, while 30% of the cases exceeded the mould count (>500 CFU g^−1^) stated in Regulation No. 461 of the Cabinet of Ministers of the Latvian Republic. Moreover, the study data on common *A. niger* and other *Aspergillus* species raise concerns related to the mycotoxigenic microbiota of commercial teas, which may affect the production of mycotoxins under improper storage conditions. The predominant *A. niger* contents indicated the potential production of OTA: Fungi were observed in 47% (*n* = 8) of the tea samples with OTA levels of 0.9–7.7 µg kg^−1^. The study data for *A. flavus* and aflatoxin indicated quantities too low to raise concerns. The presence of DON and DON secondary metabolites in commercial teas at high contamination rates suggests notable potential exposure ranging between 2.4 and 78.9% of the TDI values (1 µg kg^−1^ bw day^−1^). This study indicates that despite low contamination levels, the co-occurrence of *Aspergillus*, *Fusarium*, and *Alternaria* (ALT, ATX I) toxins should be considered when providing an assessment of these contaminants in tea beverages and evaluating the long-term effects of potential exposure to co-occurring mycotoxins.

## 5. Materials and Methods

### 5.1. Tea Samples

A total of 166 teas were purchased from tea stores and local supermarkets in Riga, Latvia, including black (*n* = 63), green (*n* = 43), oolong (*n* = 14), and Pu-erh (*n* = 20) teas imported from China, India, Japan, Taiwan, Vietnam, Sri Lanka, and Kenya, and 26 herbal teas (chamomile (*n* = 9), peppermint (*n* = 7), linden (*n* = 3), hibiscus, and rosehip (dog-rose) fruits (*n* = 7)) obtained from Ukraine, Lithuania, Poland, the Russian Federation, Latvia, and other countries. More information on these samples is described in Appendix A. The as-received teas were homogenised, placed into sample bags (Whirl-Pak^®^, Nasco, WI, USA), and stored in a dark and dry location.

### 5.2. Water Activity (a_w_)

The water activity (a_w_) of the samples was measured at 25.01 ± 0.02 °C in triplicate using an AquaLab 4TE dew-point water activity meter (Decagon Devices, Inc., Pullman, WA, USA) according to the ISO 18787:2017 standard [23].

### 5.3. Fungal Analysis

#### 5.3.1. Determination of the Total Mould Counts

The mould colony-forming unit (CFU) counts per 1 g of tea were counted according to ISO standard 21527–2:2009 [28]. A suspension of each tea sample was aseptically prepared by weighing 10 g of dry tea in a plastic filter bag supplemented with 90 mL of peptone water (0.1%) (Scharlau, Barcelona, Spain), and mixing the solution for 30 sec. Filtrates of 1 mL were spread in triplicate onto sterilised plates with dichloran 18% glycerol (DG18) Agar (Neogen, Lancashire, UK). All plates were inverted and incubated at 25 ± 0.2 °C for five days. The total fungal counts (CFU g^−1^) in the tea samples were calculated for the number of colonies on all three plates. Different dilution levels (1:10, 1:100, 1:1000, and 1:10,000) were evaluated to determine the most suitable conditions for each tea variety (for traditional teas, standard sample preparation was sufficient, whereas dilution was necessary for most herbal teas evaluated).

#### 5.3.2. MALDI-TOF-MS Analysis

Preparation of the obtained fungal isolates for MALDI-TOF MS identification was performed as previously described [63], according to Bruker’s recommendations. Identification of the observed fungal genera was performed using a Microflex LT mass spectrometer (Bruker Daltonik, Bremen, Germany) using the MALDI Biotyper software package (version 3.0) with the Filamentous Fungi Library 1.0 (Bruker Daltonik). Each sample’s mass spectrum was compared to the reference mass spectra in the Bruker MALDI-TOF fungal database (Bruker Daltonik), and the arbitrary unit score logarithmic values (ranging between 0 and 3) were determined. The Bruker interpretation criteria were used based on the consistency categories (A (species consistency), B (genus consistency), and C (no consistency)) based on the MALDI-TOF log-score values, which were interpreted as follows: scores ≥ 2.0 confirmed both the genus and species of the fungi (a high species probability was determined at scores ranging between 3.000 and 2.300), scores of ≥1.7 but <2.0 were acceptable for only genus identification, while scores < 1.7 were unreliable.

### 5.4. Mycotoxin Analysis

The multi-mycotoxin analysis was performed via the recently developed online heart-cutting 2D-LC-TOF-MS method [17]. This method was validated in the Pu-erh tea matrixes for the simultaneous determination of 70 bioactive toxins, including 42 mycotoxins produced by *Alternaria, Aspergillus, Fusarium*, and *Penicillium* strains (these compounds are summarised in Table 4).

#### 5.4.1. Standards and Reagents

The 42 tested mycotoxins were of at least 95% purity. Fusaric acid (FA), beauvericin (BEA), enniatin A (ENN A), enniatin A_1_ (ENN A_1_), enniatin B (ENN B), enniatin B_1_ (ENN B_1_), 15-acetyldeoxynivalenol (15-AcDON), 3-acetyldeoxynivalenol (3-AcDON), tentoxin (TNX), meleagrin (MEL), and citreoviridin (CVD) were purchased from Cayman Chemicals (Ann Arbor, MI, USA). Deoxynivalenol (DON), aflatoxins (AFB_1_, AFB_2_, AFG_1_, and AFG_2_), HT-2 toxin (HT-2), T-2 toxin (T-2), ochratoxin A (OTA), zearalenone (ZEN), fumonisins B_1_ and B_2_ (FB_1_, FB_2_), sterigmatocystin (STC), citrinin (CIT), fusarenon-X (FUS-X), and deoxynivalenol-3-glucoside (D3G) were all provided by Romer Labs (Tulln, Austria). Penicillic acid (PA), mycophenolic acid (MPA), and roquefortine-C (ROQ-C) were purchased from Santa Cruz Biotechnology (Dallas, TX, USA). Altenuene (ALT), alternariol (AOH), alternariol monomethyl ether (AME), altertoxin I (ATX I), ochratoxin B (OTB), aflatoxicol (AFL), verruculogen (VCL), penitrem A (PNA), neosolaniol (NEO), T-2 tetraol (T-2TET), 15-acetoxyscirpenol (15-AcS), T-2 triol (T-2TRI), and fumonisin B3 (FB3) were all purchased from Fermentek (Jerusalem, Israel). Acetonitrile (ACN), methanol (MeOH), dimethyl sulfoxide (DMSO), and ethyl acetate were all of pesticide grade and were purchased from Sigma-Aldrich (Steinheim, Germany). N,N-dimethylformamide (DMF), also of pesticide grade, and formic acid of analytical grade were both purchased from Merck Millipore (Darmstadt, Germany). Ultrapure water was generated using a Milli-Q system (Millipore, Billerica, MA, USA). The 6 mL Strata^®^ solid-phase (500 mg) extraction cartridges (55 µm, 70 Å) based on amine sorbent (NH_2_-SPE) and silica (C18-SPE) were both purchased from Phenomenex (Torrance, CA, USA).

#### 5.4.2. Sample Preparation

Sample preparation for the analysis was done via the following procedure [17]. An ethyl acetate/formic acid (99:1, *v*/*v*) extraction solution (15 mL) was added to a tea sample (2 g) weighed into a 50 mL centrifuge tube. The mixture was shaken for 15 min and then subjected to centrifugation (1500× *g*, 10 min). The extract was filtered through cellulose paper and divided into two portions. The first portion of the extract (3 mL) was evaporated until dry under a gentle nitrogen stream at 50 °C. The dry residue was then dissolved in a MeOH/water (1:1, *v*/*v*) solution (1.5 mL) and vortexed for 5 min. The second part of the extract (12 mL) was additionally cleaned via an NH_2_-SPE column that was pre-conditioned with an ethyl acetate/formic acid (99:1, *v*/*v*) mixture (5 mL) followed by loading the extract through the column and collecting the eluate in a 15 mL centrifugal tube. The column was dried by sucking out air under a vacuum for 5 min, and the eluate was evaporated under the same conditions as the first portion of the extract. The residue was dissolved in 5 mL of MeOH/water (1:1, *v*/*v*) and cleaned by passage together with the first portion of the extract through a C18-SPE column that was conditioned with a MeOH/water (1:1, *v*/*v*) solution (10 mL). After passing both parts of the extract through the C18-SPE, the column was dried under a low vacuum followed by the elution of mycotoxins with 5 mL of ACN. The eluent was evaporated, and the dry matter was dissolved in 150 µL of water/methanol (60:40, *v*/*v*) solution containing 0.1% (*v*/*v*) formic acid and transferred to autosampler vials for chromatographic analysis. Standard stock solutions of all mycotoxins were prepared in ACN, MeOH, or their compositions with DMSO, except for BEA and ENNs, which were kept in DMF (the concentration ranges for most mycotoxins are summarised in Table 1). The spiking solutions and calibration standards were prepared by the serial dilution of stock solutions and were stored in UV-protected glassware at 4 °C.

#### 5.4.3. Chromatographic Conditions

The 2D-LC analysis was performed on an UltiMate 3000 UHPLC system (Thermo Fisher Scientific, Waltham, MA, USA) containing a binary pump, a degasser, a column oven compartment, an autosampler manager fitted with a 500 μL injection loop used to capture the fraction eluted from the one-dimensional (1D) analytical column, and a programmable Rheodyne^®^ six-port switching valve. The 1D chromatographic separation was performed with a reversed-phase Kinetex C18 (1.7 µm, 10 nm, 50 × 3 mm; Phenomenex, Torrens, CA, USA) analytical column packed with fully porous silica particles. The two-dimensional (2D) separation was realised with a Luna^®^ Omega Polar C18 (3 µm, 10 nm, 100 × 3 mm; Phenomenex, USA) analytical column packed with core–shell silica particles. The column oven temperature was 40 °C. The mobile phase A used in both the 1D and 2D column gradient systems consisted of deionised water with 0.1% (*v*/*v*) formic acid. A total of 0.5 mM ammonium acetate in acetonitrile with 0.1% (*v*/*v*) formic acid was used in the gradient of the 1D column (phase B1), and acetonitrile supplemented with 0.1% (*v*/*v*) formic acid was used as the organic phase in the 2D column gradient programme (phase B2). The mobile phase flow rate was 0.35 mL min^−1^ for both dimensional columns. The elution programmes for the 1D and 2D separation and valve switching information are summarised in Table 5.

The gradient programme of the 1D column separation started with 50% of acetonitrile, providing the resolution of 16 analytes (AME, BEA, ENN A, ENN A_1_, ENN B, ENN B_1_, CVD, CIT, OTA, OTB, MPA, PNA, STC, T-2, VCL, ZEN) during a time period from 2.0 to 13.1 min. Using the same gradient system, a fraction containing 26 analytes (AFB_1_, AFB_2_, AFG_1_, AFG_2_, DON, D-3G, 15-AcDON, 3-AcDON, PA, FA, NEO, FUS-X, T-2TETR, T-2TRI, 15-AcS, FB_1_, FB_2_, FB_3_, ALT, AOH, ALX I, HT-2, TNX, AFL, ROQ-C, MEL) was collected in the sample storage loop up to 2.0 min and transferred to the 2D column.

#### 5.4.4. Mass Spectrometric Detection Conditions

An Apollo II electrospray ionisation (ESI) source and a Compact Q-ToF-MS were used for the mass analysis, performed in the positive full-scan mode and operated by the Control 4.0 software (Bruker Daltonik GmbH, Bremen, Germany). The following TOF-MS source parameters were used: capillary cap voltage of 4.5 kV, spray shield voltage of 0.5 kV, desolvation temperature of 200 °C, nitrogen flow rate of 10 L min^−1^, and nebuliser gas flow pressure of 2 bar. The Bruker software HyStar 3.2 and QuantAnalysis 4.3 were used for data acquisition and analysis, respectively. The instrumental parameters of the mycotoxin analysis are provided in Table 4.

#### 5.4.5. Validation of the Method

Validation was performed according to the performance guidelines of the Commission Regulation (EU) No 519/2014 and the “Guidance Document on the Estimation of LOD and LOQ for Measurements in the Field of Contaminants in Feed and Food” [23]. The selectivity, trueness, linearity, repeatability expressed as the relative standard deviation (RSD, %), and the uncertainty of the method (MU, %) developed in this study were verified according to the criteria described in these documents using blank samples spiked with mycotoxin standards. To evaluate the linearity, five-point calibration curves were constructed to calculate the determination coefficients (R^2^). The precision, repeatability, and recovery studies during intraday tests (two days within a one-week period) were performed by analysing six replicates at each of the three determined spiking levels. The mean values of each replicate were compared to the acceptance criteria specified by the EC regulation No. 519/2014. The signal-to-noise (S/N) approach was used to estimate the LOD and the LOQ. The chromatographic noise and analytical response were estimated using chromatograms of the spiked samples. The LOD and LOQ were defined based on signal (S)-to-noise (N) ratios of S/N > 3 and S/N > 10, respectively. The measurement uncertainties (MU) were calculated according to Eurachem/CITAC Guide CG 4 [22]:(1)$$MU=k\times \sqrt{u^{\prime }{\left(wR\right)}^{2}+\;u^{\prime }{\left(bias\right)}^{2}}$$
where *u′(wR)* is the within-laboratory reproducibility, *u′(bias)* is the uncertainty component arising from the method and laboratory bias, *MU* is the expanded measurement uncertainty, and the coverage factor *k* = 2 is at a 95% confidence level [22].

### 5.5. Dietary Exposure Assessment

The methodology reported in our previous study was also used here [16]. Consumption data provided from surveys conducted between 2012 and 2014 were applied for the initial evaluation, and the following average daily intake values of tea beverages (mL) were included in the calculations: 318 mL of black tea and 345 mL of green tea; peppermint infusions: 251 mL [18]. Unfortunately, no data on Pu-erh tea were available. We assumed that the intake levels of Pu-erh are the same as the volumes found for fermented black tea. The data were selected to fit the necessary significance of the average values, with more than 10 observations required for short-term consumption (acute exposure), which was in line with the recommendations for the use of these statistics to assess dietary exposure to hazardous substances [64].

For the acute risk assessment, the probable daily intake (PDI) values were calculated for two scenarios—exposure to the highest contamination level and the upper-bound (UB) contamination levels. A conservative approach was applied to the occurrence data and all cases of non-detection were replaced with the LOQ values for the particular analytes, in order to obtain the UB values according to EFSA recommendations for the dietary exposure evaluation methodology [65].

As the current study did not provide an evaluation of beverages but only raw tea samples, the worst-case scenario was assessed under the assumption that all of the mycotoxin content found in the dry teas would be transferred to the tea infusions. Our recent study indicated that the DON and ZEN levels in dry herbs and their infusions can be transferred almost fully into beverages [16]. However, the individual solubility in water should be considered. For example, individual solubility in water for DON can reach 55 mg mL^−1^, according to the data reported in the literature [2]. In the two considered scenarios, it was assumed that the consumed tea beverage contained transferred mycotoxins at a level equal to the mean or the maximum concentrations found in the analysed tea samples. The PDI values were calculated using the following equation:(2)$$PDI=C\times m_{p}\times V_{c}/\left(V_{p}\times B\right),$$
where *C* is the found mean or highest (maximum) mycotoxin concentration found for the analysed dry tea samples of the selected tea variety, *m_p_* is the assumed tea weight per teaspoon (2.00 g), and *V_p_* is the volume of a typical tea beverage portion (200 mL), infused for 5 min according to common data and tea manufacturer recommendations; *V_c_* is the medium intake volume (mL) of tea beverages by the target group, and *B* is the median selected body weight of 70 kg for adolescents, adults, and the elderly in Europe, which was adopted based on the EFSA recommendations [66].

Most of the bagged tea samples included in the current study contained 1.0 to 2.0 g of tea. The maximum sample weight per portion was chosen to be 2.0 g, which fit the approximate weight of one teaspoon of *Camelia sinensis* tea. This coincided with results reported by other researchers, namely that 1.5 to 2.0 g of dry tea and 150–250 mL of water are commonly used for the preparation of tea beverages [2,14,15,16].

The PDIs were compared with the recommended tolerable daily intake (TDI) levels found by the EFSA for certain mycotoxins: 1 µg kg^−1^ bw day^−1^ of DON and also the sum of DON and its derivatives (e.g., D3G, 3-ACDON, and 15-AcDON), 17.1 µg kg^−1^ bw day^−1^ of OTA (which corresponds to the tolerable weekly intake (TWI) of 120 µg kg^−1^ bw week^−1^), and 0.25 µg kg^−1^ bw day^−1^ of ZEN [67,68]. The average human body weight of 70 kg according to the EFSA recommendations [69] was used.

Due to the high prevalence of AFB_1_ and 4AFs determined in *C. sinensis* and peppermint teas, their PDI values were also evaluated. However, due to the carcinogenic effects, AFs have no tolerable levels. To estimate the potential health risk related to the consumption of teas contaminated with AFs, the benchmark dose (BMD) approach of the European Food Safety Authority was used. For dose–response modelling of aflatoxins, the BMD lower confidence limit (BMDL_10_) for a 10% increase in cancer incidence obtained from animal study data modelling (170 ng kg^−1^ bw day^−1^) was considered (European Food Safety Authority [70]. This method was reported in a recent study, whereas the margin of exposure (MOE) using the benchmark dose (BMD) approach of the European Food Safety Authority was used [45].

## Figures and Tables

**Figure 1 toxins-12-00555-f001:**
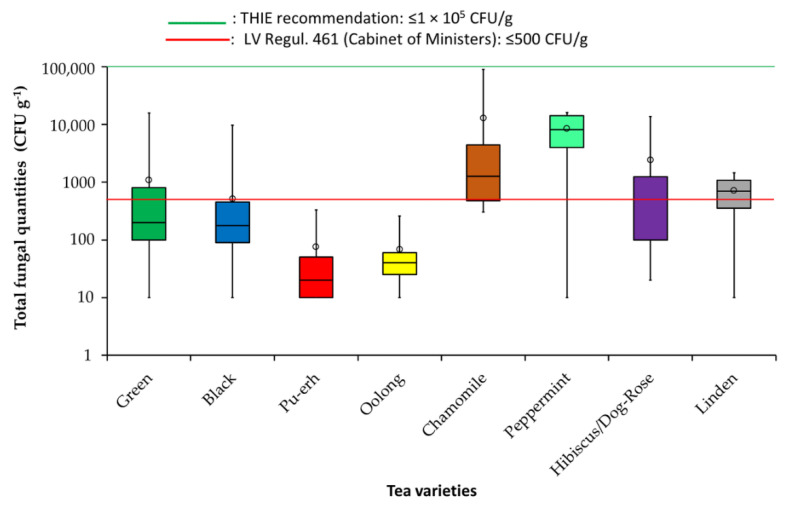
The range of the total fungal colony-forming unit counts per gram of tea and their mean levels (unfilled round markers). The upper line (green) indicates the quality level recommended by the THIE organisation, while the lower line (red) shows the recommended maximum levels of total fungi and yeasts in commercial teas according to Regulation No. 461 of the Cabinet of Ministers (Republic of Latvia) [29].

**Figure 2 toxins-12-00555-f002:**
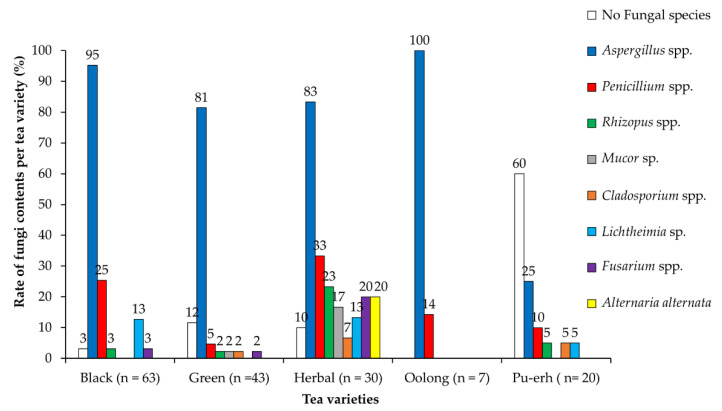
Number of contaminated tea samples with predominant fungi genera.

**Figure 3 toxins-12-00555-f003:**
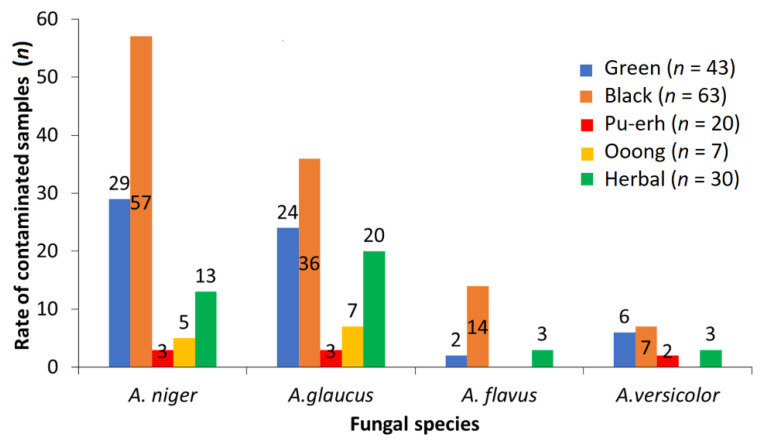
Distribution of the four most prevailing fungal species in different tea varieties.

**Figure 4 toxins-12-00555-f004:**
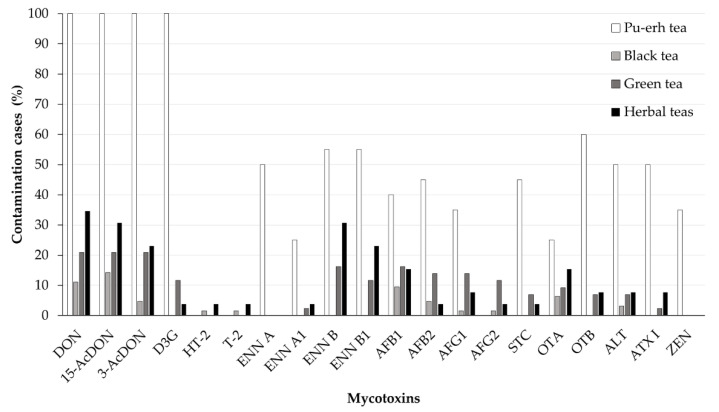
Summary of the distributions of individual mycotoxins in the samples of different tea varieties.

**Figure 5 toxins-12-00555-f005:**
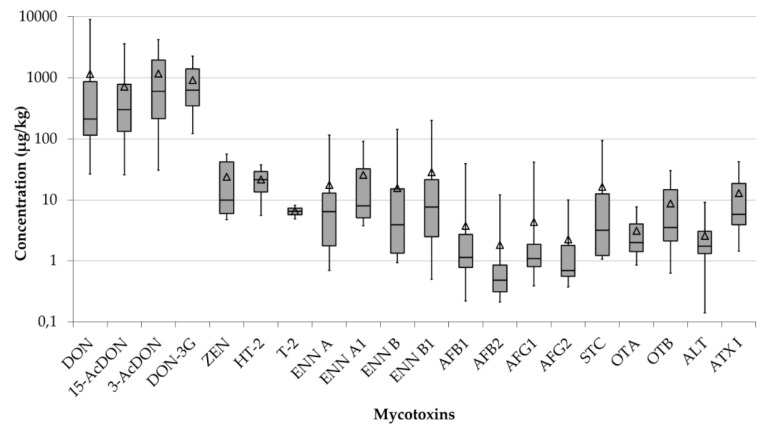
The range of individual mycotoxins’ concentrations and the mean levels (unfilled triangle markers) determined in the 70 mycotoxin-positive tea samples (concentrations given on a logarithmic scale).

**Table 1 toxins-12-00555-t001:** Performance of the method for mycotoxin analysis.

Mycotoxin	Linear Range (µg kg^−1^)	Linearity (R^2^)	LOD (µg kg^−1^)	LOQ (µg kg^−1^)	Recovery (%) (*n* = 36)	Intraday RSD (%) (*n* = 36)	MU (%)
AFB_1_	1.0–25	0.998	0.05	0.15	96	9	18
AFB_2_	0.25–5	0.999	0.07	0.24	90	10	19
AFG_1_	1.0–25	0.998	0.05	0.18	92	9	17
AFG_2_	0.25–5	0.994	0.08	0.27	93	8	15
ALT	1.0–25	0.9997	0.3	1.01	92	10	19
ATX I	1.0–25	0.9995	0.13	0.45	95	9	18
D3G	25–500	0.999	1.9	6.3	96	14	28
DON	25–500	0.996	1.55	5.15	108	6	11
ENN A	1.0–25	0.9993	0.04	0.12	103	8	16
ENN A_1_	1.0–25	0.9997	0.03	0.11	103	7	13
ENN B	1.0–25	0.9992	0.05	0.16	102	8	16
ENN B_1_	1.0–25	0.9998	0.08	0.27	97	4	9
HT-2	5.0–100	0.9996	1.22	4.07	97	8	16
OTA	1.0–25	0.9993	0.07	0.24	93	6	13
OTB	1.0–25	0.999	0.04	0.13	90	6	13
STC	1.0–25	0.9996	0.08	0.26	104	8	17
T-2	5.0–100	0.9992	1.22	4.07	94	7	14
ZEN	5.0–100	0.9993	0.69	2.29	96	8	16
15-AcDON	25–500	0.998	5.77	19.2	96	7	13
3-AcDON	25–500	0.994	4.23	14.2	95	6	13

Notes: ALT—altenuene, ATX I—altertoxin I, D3G—deoxynivalenol-3-glucoside, HT-2—HT-2 toxin, OTB—ochratoxin B, STC—sterigmatocystin, T-2—T-2 toxin, ZEN—zearalenone, 15-AcDON—15-acetyldeoxynivalenol, 3-AcDON—3-acetyldeoxynivalenol, D3G—deoxinyvalenol-3-glucoside, LOD—limit of detection, LOQ—limit of quantification, Intraday RSD—the intraday precision estimated by calculating the relative standard deviation of parallel measurements, MU—measurement uncertainty.

**Table 2 toxins-12-00555-t002:** Characterisation of individual fungal species determined in tea samples.

Fungal Prevalence	Indicated Fungal Species (*n* = Positive Samples)
67%	*A. niger* (*n* = 105)
57%	*A. glaucus* (*n* = 87)
10.0–11.9%	*P. chrysogenum* (*n* = 16), *A. versicolor* (*n* = 18), *A. flavus* (*n* = 19)
5.0–7.5%	*F. incarnatium* (*n* = 8), *Rhizopus oryzae* (*n* = 9), *A*. *nidulans* (10), *A. amstelodami* (*n* = 12), *Lichthemia corymbifera* (*n* = 12)
2.5–4.9%	*P. discolor* (*n* = 4), *Cladosporium chrysogenum* (*n* = 4), *Paecilomyces variotii* (*n* = 4), *Alternaria alternata* (*n* = 5), *Mucor circinelloides* (*n* = 6), *Phoma sorghina* (*n* = 6)
0.6–1.9%	*P. roqueforti* (*n* = 1), *P. italicum* (*n* = 1), *Rhizopus stolonifera* (*n* = 1), *Chaetomium globosum* (*n* = 1), *Fennelia flavipes* (*n* = 1), *Aureobasidium pullulans* (*n* = 1), *Trichoderma koningii* (*n* = 1), *A.oryzae* (*n* = 2), *P. digitatum* (*n* = 2), *P. corylophilum* (*n* = 2), *F. equiseti* (*n* = 2), *Cladosporium chrysogenum* (*n* = 2), *P. citrininum* (*n* = 3), *Trichophyton mentagrophytes* (*n* = 3), *Scopulariopsis brevicaulis* (*n* = 3), *Curvularia pallescens* (*n* = 3)

Notes: A.—*Aspergillus*, F.—*Fusarium*, P.—*Penicillium* species.

**Table 3 toxins-12-00555-t003:** Acute exposure risk assessment of the DON metabolites, OTA, and ZEN, with full mycotoxin extraction into the beverages (worst-case scenario).

Tea variety	Mycotoxin	Concentration (µg kg^−1^)		PDI (µg kg^−1^ bw day^−1^)		PDI/TDI (%) ***	
		UB	Max	UB	Max	UB	Max
Green	OTA	0.60	3.40	2.96 × 10^−5^	1.68 × 10^−4^	0.17	0.98
	AFB_1_	0.47	2.99	2.32 × 10^−5^	1.47 × 10^−4^	0.01	0.09
	∑(AFs) *	1.55	7.67	7.64 × 10^−5^	3.78 × 10^−4^	0.04	0.22
	DON	127	859	6.26 × 10^−3^	4.23 × 10^−2^	0.63	4.23
	∑(DON derivatives) **	740	4512	3.65 × 10^−2^	2.22 × 10^−1^	3.60	22.2
Black	OTA	0.97	7.70	4.41 × 10^−5^	3.50 × 10^−4^	0.26	2.05
	AFB_1_	2.36	39	1.07 × 10^−4^	1.77 × 10^−3^	0.06	1.04
	∑(AFs) *	6.35	103	2.88 × 10^−4^	4.68 × 10^−3^	0.17	2.75
	DON	28.5	211	1.29 × 10^−3^	9.59 × 10^−3^	0.13	0.96
	∑(DON derivatives) **	103	549	4.68 × 10^−3^	2.49 × 10^−2^	1.00	2.50
Pu-erh	OTA	0.71	4.05	1.84 × 10^−5^	1.86 × 10^−4^	0.19	1.08
	AFB_1_	1.30	9.17	5.91 × 10^−5^	4.17 × 10^−4^	0.03	0.25
	∑(AFs) *	3.18	20.1	1.44 × 10^−4^	9.13 × 10^−4^	0.08	0.54
	DON	2221	8946	4.06 × 10^−1^	4.06 × 10^−1^	10.1	40.6
	∑(DON derivatives) **	5461	17360	7.89 × 10^−1^	7.89 × 10^−1^	24.8	78.9
	ZEN	11.3	56.1	5.13 × 10^−4^	2.76 × 10^−3^	0.20	1.11
Peppermint	OTA	2.99	4.2	1.07 × 10^−4^	1.51 × 10^−4^	0.63	0.88
	AFB_1_	2.84	5.62	1.02 × 10^−4^	2.02 × 10^−4^	0.06	0.12
	∑(AFs) *	8.16	15.7	2.93 × 10^−4^	5.63 × 10^−4^	0.17	0.33
	DON	567	953	7.67 × 10^−2^	3.42 × 10^−2^	2.03	3.42
	∑(DON derivatives) **	3599	5631	1.29 × 10^−1^	2.02 × 10^−1^	12.9	27.7

UB: upper-bound level of mean. * ∑(AFs): the total concentration of AFB_1_, AFB_2_, AFG_1_, and AFG_2_. ** ∑(DON derivatives): the total concentration of DON, D3G, 3-AcDON, and 15-AcDON; TDI (DON) = 1 µg kg^−1^ bw day^−1^, TDI(∑(DON metabolites)) = 1; TDI(OTA) = 0.0171 µg kg^−1^ bw day^−1^; TDI(ZEN) = 0.25 µg kg^−1^ bw day^−1^. *** For sum of aflatoxins, the benchmark dose lower confidence limit (BMDL_10_) for a 10% increase in cancer incidence obtained from animal study data modelling (170 ng kg^−1^ bw day^−1^) was considered.

**Table 4 toxins-12-00555-t004:** Instrumental parameters of the mycotoxin analysis in teas.

Mycotoxin	RT (min)	Quantification Ion			Qualifier Ion		
		Theoretical Mass (*m/z*)	Measured Mass (*m/z*)	Mass Accuracy (ppm)	Theoretical Mass (*m/z*)	Measured Mass (*m/z*)	Mass Accuracy (ppm)
AOH *	2.3	259.0601	259.0598	1.16	281.2156	281.2144	4.27
T-2TET ***	2.4	321.1308	321.1301	2.18	299.1489	299.1478	3.68
CIT ****	2.7	251.0914	251.091	1.59	273.0733	273.0728	1.83
MPA ****	2.8	321.1332	321.1328	1.24	338.1598	338.1588	2.95
OTB **	3	370.1285	370.1279	1.62	392.1104	392.1089	3.83
CVD ****	3.3	403.2115	403.2099	3.97	425.1934	425.191	3.29
T-2 ***	3.6	484.2541	484.2528	2.68	489.2095	489.208	3.07
D3G ***	4.5	476.2126	476.2114	2.52	459.186	459.1851	1.96
DON ***	4.7	297.1333	297.1338	−1.68	319.1152	319.1161	−2.82
FA ***	4.8	180.1019	180.1015	−2.22	197.1284	197.12	−2.03
FUS-X ***	4.8	372.1652	372.1642	2.67	355.1387	355.1375	3.38
15-AcDON ***	4.9	356.1703	356.1709	−1.68	339.1438	339.1449	−3.24
NEO ***	4.9	400.1965	400.1951	3.5	405.1519	405.1503	3.95
15-AcS ***	5	342.1646	342.1659	−3.8	347.1465	347.1473	−2.3
PA ****	5	171.0651	171.0649	1.17	193.0471	193.0466	2.59
3-AcDON ***	5.1	339.1438	339.1431	2.06	356.1703	356.1697	1.68
ZEN ***	5.6	319.154	319.1548	−2.51	341.1359	341.1364	−1.47
OTA **	5.8	404.0895	404.089	1.24	426.0715	426.0704	2.58
AME *	5.9	273.0757	273.0747	3.66	295.0577	295.0565	4.07
STC **	6.9	325.0707	325.0712	−1.54	347.0526	347.0529	−0.86
VCL ****	7.7	534.221	534.2215	−0.94	512.2391	512.2401	−1.95
PNA ****	10.7	634.2929	634.293	−0.16	656.2749	656.2751	−0.3
ENN B ***	11.5	657.4433	657.4444	−1.67	662.3987	662.3993	−0.91
ENN B_1_ ***	11.8	671.4589	671.4584	0.74	676.4143	676.4153	−1.48
BEA ***	11.9	801.4431	801.4245	−2.75	806.3987	806.3999	−1.49
ENN A ***	12.1	685.4746	685.4732	2.04	690.43	690.4289	1.59
ENN A_1_ ***	12.3	699.4902	699.4884	2.57	704.4456	704.4435	2.98
MEL ****	13	434.1822	434.1807	3.45	456.1657	456.1673	−3.51
ALT *	13.1	293.1019	293.1009	3.41	315.0839	315.0827	3.81
T-2TRI ***	13.1	400.2329	400.2317	3	383.2064	383.205	3.65
ROQ-C ****	13.2	390.192	390.1903	4.37	407.2186	407.217	3.93
FB_1_ ***	13.3	722.3957	722.3987	−4.15	744.3776	744.3778	−0.27
AFG_1_ **	13.4	331.0812	331.0808	1.21	353.0632	353.0623	2.55
AFB_1_ **	13.5	315.0863	315.0852	3.49	337.0682	337.0673	2.67
AFG_2_ **	13.6	331.0812	331.0808	1.21	353.0632	353.0623	2.55
AFB_2_ **	13.7	315.0863	315.0852	3.49	337.0682	337.0673	2.67
FB_3_ ***	13.7	706.4008	706.3981	3.82	728.2827	728.2837	−1.37
HT-2 ***	13.7	442.2425	442.2428	−0.68	447.1989	447.1975	3.13
AFL **	13.8	337.0682	337.0669	3.87	315.0863	315.0852	3.49
TNX *	13.8	415.2339	415.2355	−3.85	432.2605	432.2616	−2.54
ALX I *	13.9	370.1285	370.127	4.05	353.1019	353.1006	3.68
FB_2_ ***	14	706.4006	706.4011	−0.71	728.3828	728.3809	2.61

RT: retention time. Origins of fungal genera: *—*Alternaria* spp., **—*Aspergillus* spp., ***—*Fusarium* spp., **** —*Penicillium* spp.

**Table 5 toxins-12-00555-t005:** Gradient eluting programme and six-port valve-switching information.

^1^D-LC Separation			^2^D-LC Separation			Valve Position
Time (min)	A (%)	B_1_ (%)	Time (min)	A (%)	B_2_ (%)	
0	50	50	0	45	55	6-1
0.5	50	50	0.5	45	55	1-2
1.9	50	50	1.9	45	55	
3.0	50	50	2.7	45	55	6-1
8.0	2	98	3.0	90	10	
11	2	98	5.0	90	10	
11.5	90	10	9.0	2	98	
13	90	10	11	2	98	
13.5	60	40	11.2	45	55	
16	60	40	16	45	55	

A: deionised water containing 0.1% (*v*/*v*) of formic acid, B_1_: 0.5 mM ammonium acetate in acetonitrile containing 0.1% (*v*/*v*) of formic acid, B_2_: acetonitrile containing 0.1% (*v*/*v*) of formic acid.

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
