# Peer review of "Determination of Fungi and Multi-Class Mycotoxins in Camelia sinensis and Herbal Teas and Dietary Exposure Assessment"

_toxins, 2020, doi:10.3390/toxins12090555_

Round 1
Reviewer 1 Report
The manuscript “Determination of Fungi and Multi-Class Mycotoxins in Camellia Sinensis and Herbal Teas and Dietary Exposure Assessment” dwells upon contamination of traditional black and green tea, Pu-erh, oolong and herbal tea with micromycetes and mycotoxins. The results are well presented and discussed. The quantity of samples analyzed is sufficient for the dietary exposure evaluation. In my opinion, this article will be interesting for the scientific community not only as a basis for risk assessment but as an example of a comprehensive approach of fungal-mycotoxin contamination analysis.
However, there are several questions:
Materials and methods (Paragraph 5):
- The original method 2dLC-MS method has been validated for Pu-erh tea. In the present manuscript, there is no discussion on its validation or applicability for mycotoxins determination in other studied Camellia sinensis tea types (black, green, oolong) and herbal tea. I suppose that peppermint and chamomile matrixes differ from those of Pu-erhs.
- Description of HPLC conditions (Paragraph 5.4.3) is not clear.
- I suggest adding information on the time for capturing polar mycotoxins fraction from the 1D column to the loop.
- As there are at least two 6-port valves (switching the loop and flow to MS) it is desirable to specify which one is mentioned in Table 5.
- It is interesting and to my mind useful for other scientists engaged in method development to specify 1D or 2D column each mycotoxin is eluted from (if it is possible)
- Sample preparation (Paragraph 5.4.2). It seems not reasonable to divide extract into two parts to subject its major part (80%) to NH2-SPE clean-up and leave the minor part untreated. Please, explain.
Dietary exposure (Paragraph 2.4):
- The mean concentrations of mycotoxins in various tea kinds were calculated, taking into account only mycotoxin-positive samples. It seems inapplicable for the risk assessment. Negative samples should be included using the UB or LB approach.
- AFLs occurrence was about 30% according to the presented data. It seems interesting to estimate daily intake of AFLB1 and/or total AFLs. As far as no TDI can be set for AFLs, PDI can be compared to chronic dietary exposure [Risk assessment of aflatoxins in food, doi: 10.2903/j.efsa.2020.6040].
Several mistyping were also noted (like omitted word (lines 92-93); STG (line 280), italics for genera names should be checked (e.g. lines 661, 665…)
Thus, I suggest some corrections and comments before the article can be published in Toxins.

Author Response
We would like to thank the Reviewer 1 for valuable and constructive commentaries and suggestions. The Reviewer has precisely indicated the aim of this paper related to an approach of fungal-mycotoxin contamination analysis. The manuscript was re-checked and revised; were it was necessary. Changes made in the manuscript are marked using track changes mode.
Here are the answers to the first Reviewers commentaries and questions:
However, there are several questions:
Materials and methods (Paragraph 5):
- The original method 2dLC-MS method has been validated for Pu-erh tea. In the present manuscript, there is no discussion on its validation or applicability for mycotoxins determination in other studied Camellia sinensis tea types (black, green, oolong) and herbal tea. I suppose that peppermint and chamomile matrixes differ from those of Pu-erhs.
Authors’ response: The applicability of the method was proved by quality control (QC). QC samples were included in every analysis sequence, as well as using for quantification matrix-matched calibration on blank sample, that included mix of Camellia sinensis teas, and herbal tea blend. The accuracy of QC values for all the compounds ranged between 70% and 124%. And the linearity of matrix-matched calibration curves for all the target compounds was quite similar to values reached by the original method validated on the Pu-erh teas (Bogdanova et al., 2020, https://doi.org/10.1016/j.chroma.2020.461145).
We have added additional information in the manuscript (Lines 120-125, pages 3,4): “The applicability of the method was proved by quality control (QC). QC samples were included in every analysis sequence, as well as using for quantification matrix-matched calibration on blank sample, that included mix of C. sinensis teas, and herbal tea blend. The accuracy of QC values for all the compounds ranged between 70% and 124%.
The linearity of matrix-matched calibration curves for all the target compounds was quite similar to values reached by the original method validated on the Pu-erh teas [17].".
- Description of HPLC conditions (Paragraph 5.4.3) is not clear.
- I suggest adding information on the time for capturing polar mycotoxins fraction from the 1D column to the loop.
Authors’response; please see belo the answer to the third question of Paragraph 5.4.3.
- As there are at least two 6-port valves (switching the loop and flow to MS) it is desirable to specify which one is mentioned in Table 5.
Authors’response: There was only one 6-port valve used for switching flow direction (which is also the one mentioned in Table 5), that let us to control, which fraction of sample was collected in the sample loop and which one was directed to MS.
- It is interesting and to my mind useful for other scientists engaged in method development to specify 1D or 2D column each mycotoxin is eluted from (if it is possible)
Authors’ response: Additional information was added to page 24 (lines 1173-1178): “The gradient program of 1D column separation started with 50% of acetonitrile, providing the resolution of 16 analytes (AME, BEA, ENN A, ENN A1, ENN B, ENN B1, CVD, CIT, OTA, OTB, MPA, PNA, STC, T-2, VCL, ZEN) during time period from 2.0 to 13.1 min. Using the same gradient system, a fraction containing 26 analytes (AFB1, AFB2, AFG1, AFG2, DON, D-3G, 15-AcDON, 3-AcDON, PA, FA, NEO, FUS-X, T-2TETR, T-2TRI, 15-AcS, FB1, FB2, FB3, ALT, AOH, ALX I, HT-2, TNX, AFL, ROQ-C, MEL) was collected in the sample storage loop up to 2.0 min and transferred to the 2D column.“
- Sample preparation (Paragraph 5.4.2). It seems not reasonable to divide extract into two parts to subject its major part (80%) to NH2-SPE clean-up and leave the minor part untreated. Please, explain.
Authors response’: An additional filtration step of the extracts through NH2-SPE column resulted in chromatographic signal deterioration for ENNs and BEA, as well as the determination of FBs became impossible because of their binding to the sorbent material, which has been also mentioned in an earlier studies. On the other hand, this step improved the determination of other compounds by elimination of undesirable matrix components. So, the best solution to improve the determination of the most mycotoxins without significant affecting the ENNs, BEA and FBs determination, is to pass one part of the extract through NH2-SPE column and leave the rest of extract without additional clean up. During experiments, it was found that the volume of 3 mL is optimal that is needed for filtration.
Dietary exposure (Paragraph 2.4):
- The mean concentrations of mycotoxins in various tea kinds were calculated, taking into account only mycotoxin-positive samples. It seems inapplicable for the risk assessment. Negative samples should be included using the UB or LB approach.
Authors’ response: changes in the section 5.5. Dietary exposure assessment (lines 1221-1224): text “For the acute risk assessment, the probable daily intake (PDI) values were calculated for two scenarios—exposure to the highest contamination level and the upper-bound (UB) contamination levels. A conservative approach was applied to the occurrence data and all cases of non-detection were replaced with the LOQ values for the particular analytes, in order to obtain the UB values according to EFSA recommendations for dietary exposure evaluation methodology [70].” was added, that indicates that all the calculations of the mean in Table 3 (Table 3. Acute exposure risk assessment of the DON metabolites, OTA and ZEN, with full mycotoxin extraction into the beverages (worst-case scenario) were replaced by upper-bound (Mean + 95% confidence level). Data of the Paragraph 2.4 were rechecked and recalculated.
- AFs occurrence was about 30% according to the presented data. It seems interesting to estimate daily intake of AFB1 and/or total AFLs. As far as no TDI can be set for AFLs, PDI can be compared to chronic dietary exposure [Risk assessment of aflatoxins in food, doi: 10.2903/j.efsa.2020.6040].
Authors’ response: the risk assessment part was extended by data of AFB1 and summary AF exposure assessment (calculation of PDI values). The occurrence assessment was based on the procedure reported in a recent study [10.3389/fmicb.2020.00446] and notes to that were included in Paragraph 2.4 and in Paragraph 5.5 (lines 1253-1261)
Several mistyping were also noted (like omitted word (lines 92-93); STG (line 280), italics for genera names should be checked (e.g. lines 661, 665…)
Authors’ response: Lines 92-93 were checked. STG changed to STC (286), changes of genera names (A. niger, C. sinensis, etc.) into italics were provided within whole manuscript, changes are presented by tracking function.
Reviewer 2 Report
I have some recommendations or comments:
It is very difficult to know if any mycotoxin content is because of bad harvesting and following storage or if it is given by bad storage conditions of final product. Did you find any correlation between the content of mycotoxins in teas in the same harvest and producer country and different harvest and producer country?
Chapter 5.5 Dietary exposure assessment: "Consumption data provided from recent surveys conducted between 2012 and 2014..." Now, it is year 2020, it is not recent surveyes/study.
It is huge data you obtained. What about any project support. You didnt mention about that.
Author Response
The authors thank the 2nd Reviewer for the positive evaluation of the article, constructive suggestions.
Responses to the commentaries and questions are shown below each note of the Reviewer. Changes made in the manuscript are marked using track changes mode.
- It is very difficult to know if any mycotoxin content is because of bad harvesting and following storage or if it is given by bad storage conditions of final product. Did you find any correlation between the content of mycotoxins in teas in the same harvest and producer country and different harvest and producer country?
Authors’ response: As already pointed out by the Reviewer, it is very difficult to assess the correlation due to many unknown factors about the origin of the samples. Only some of the teas that were weighed were known to be directly exported with some information, but most packaged teas (including loose-leaf or tea-bags) often did not even have a clear origin for the type of tea, and were often found to contain other additives. It was found that there are many unknowns and a lack of data on the production process, especially for teas that were not obtained from tea houses (loose leaf tea samples) but were purchased in supermarkets. Thus, the screening results showed more trends in contamination and mycotoxin distribution levels. For example, Pu-erh teas and herbal teas had higher levels of mycotoxin contamination and multi-mycotoxin content (especially emerging mycotoxins - ENNs, Alternaria toxins). In the section on contamination levels and in the discussion section, the authors tried to describe the trends as well as to compare the contamination levels in different tea varieties. Direct correlation was not found between countries, as well as the type of tea packaging due to non-determined factors such as growing conditions, harvest season, processing technology, storage and transport conditions.
- Chapter 5.5 Dietary exposure assessment: "Consumption data provided from recent surveys conducted between 2012 and 2014..." Now, it is year 2020, it is not recent surveyes/study.
Authors’ response: The term “recent” was deleted. Our colleagues have also conducted a more recent survey, but unfortunately these data are still being analysed, so it was not possible to include them and we relied on the currently known Latvian consumer survey data on tea consumption.
- It is huge data you obtained. What about any project support? You didn’t mention about that.
Authors’ response: This study was provided within Latvian “Post-doctoral Research Aid” funded from the European Regional Development Fund (ERDF) grant No. 1.1.1.2/VIAA/1/16/219.
Additional sections such as "Funding" and "Acknowledgments" were included in the revised manuscript, which noted acknowledgements to financial support from ERDF post-doctoral grant and the MDPI Language editing service for professional and diligent proofreading of this manuscript.